# TQL: Scaling Q-Functions with Transformers by Preventing Attention Collapse

Perry Dong [*]  Kuo-Han Hung [*]  Alexander Swerdlow  Dorsa Sadigh  Chelsea Finn

Stanford University

https://pd-perry.github.io/transformer-q-learning/

## Abstract

Despite scale driving substantial recent advancements in machine learning, reinforcement learning (RL) methods still primarily use small value functions. Naively scaling value functions – including with a transformer architecture, which is known to be highly scalable – often results in learning instability and *worse* performance. In this work, we ask what prevents transformers from scaling effectively for value functions? Through empirical analysis, we identify the critical failure mode in this scaling: attention scores collapse as capacity increases. Our key insight is that we can effectively prevent this collapse and stabilize training by controlling the entropy of the attention scores, thereby enabling the use of larger models. To this end, we propose Transformer Q-Learning (TQL), a method that unlocks the scaling potential of transformers in learning value functions in RL. Our approach yields up to a 43% improvement in performance when scaling from the smallest to the largest network sizes, while prior methods suffer from performance degradation.

## 1. Introduction

Recent advances in deep learning have demonstrated that scaling model capacity yields substantial performance gains across domains such as natural language processing (OpenAI et al., 2024), computer vision (Siméoni et al., 2025), and robotics (Physical Intelligence et al., 2025b; Team et al., 2025). However, unlike supervised learning—where increasing parameters typically improves performance—reinforcement learning (RL) has historically faced significant scaling challenges. Larger networks often fail to better capture the data distribution and instead induce training instabilities that degrade performance. Recent works have explored using larger policy networks in RL. In this work, we focus on scaling up value functions. As policies are ultimately derived from value functions, increased parameterization of the value function should substantially benefit RL performance; yet, using larger value function networks often results in *worse* performance, and how to effectively scale them remains an open question. In traditional supervised learning, transformers have been demonstrated to scale well across numerous domains. Therefore, we ask: what prevents transformers from effectively scaling in RL value learning?

Popular prior approaches to scaling up networks in RL employ periodic network resets to maintain plasticity under higher update-to-data (UTD) ratios during online training (Nauman et al., 2024; Schwarzer et al., 2023). While these methods preserve learning capacity, they are compute-heavy as the network needs to be retrained from scratch and they risk catastrophic forgetting through parameter reinitialization that can make online learning unsafe. Alternative approaches have explored normalization techniques (Lee et al., 2025) or alternative architectures (Obando-Ceron et al., 2024), which often require specific architectural changes, making it challenging to apply to all settings such as in the case of pretrained networks. In this work, we focus on understanding why transformers do not scale for value function learning in RL, and propose a general, minimal framework to address these issues and enable scaling.

To this end, we propose Transformer Q-Learning (TQL), a simple framework for scalable value function training in RL. TQL leverages the scalability of the transformer architecture for value function training, while keeping architectural changes as minimal as possible. Directly applying transformers to value learning yields severe pathologies (Section 5.2). Through empirical analysis, we identify the critical failure mode in scaling up transformer value function training: transformers collapse their attention weights as capacity increases, attending predominantly to a handful

---

[*]Equal contribution  Correspondence to: Perry Dong <perryd@stanford.edu>, Kuo-Han Hung <khhung@stanford.edu>.

*Proceedings of the $43^{rd}$ International Conference on Machine Learning*, Seoul, South Korea. PMLR 306, 2026. Copyright 2026 by the author(s).

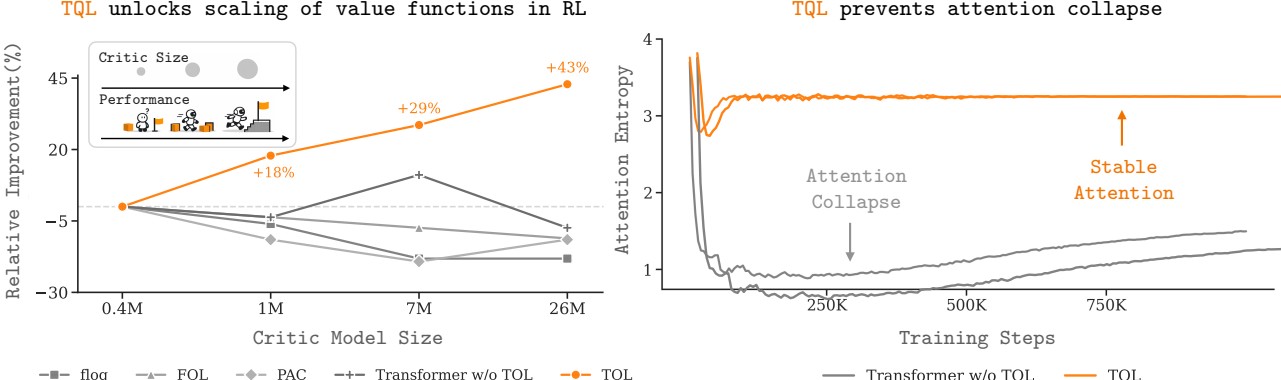

*Figure 1.* **TQL unlocks scaling of value functions in RL by preventing attention collapse.** Left: Scaling results of TQL compared with prior approaches with different generative model backbones. The results are reported as relative improvement over the smallest model size, averaged across seeds and all 25 tasks in our evaluation suite. TQL increases in performance as network size increases, while prior approaches are not able to effectively use the extra capacity and drop in performance. Right: Attention entropy with and without TQL, averaged across 25 tasks. TQL scales effectively by preventing attention collapse that occurs with scaling up the transformer architecture for value function training.

of tokens while ignoring the rest, and this results in larger models struggling to learn appropriate attention patterns during value bootstrapping. As a simple remedy, we propose per-layer control of the entropy of the attention scores. By adjusting the entropy of attention scores toward a target value, TQL ensures that the model effectively distributes attention across all input tokens to enable stable training by preventing collapse. This enables the larger model to use its capacity to fit useful signals and unlock scaling of transformers for value function training (Figure 1).

Our main contribution is TQL, a framework for guiding the attention entropy as a way to provide stable, effective value function training as the model scales up in capacity. We analyze TQL in an offline RL setting and evaluate on challenging continuous control tasks from the OGBench benchmark (Park et al., 2025a), demonstrating the effectiveness of TQL in scaling up value functions compared to prior approaches.

## 2. Related Work

**Offline RL.** The goal of offline RL (Levine et al., 2020) is to train a policy on an offline dataset without environment interactions. The central concern of offline RL is handling actions that are out of the dataset distribution, where estimates of value are unreliable. There are two primary approaches to address this issue. One approach is to use conservatism to learn a pessimistic value function to penalize out-of-distribution actions (Tarasov et al., 2023; Kumar et al., 2020; Wu et al., 2019; 2021; Yu et al., 2020). Another is to apply policy constraints to keep the policy actions close to the behavior distribution (Dong et al., 2025d; Park et al., 2025b; Hansen-Estruch et al., 2023; Kostrikov et al., 2021; Fujimoto & Gu, 2021; Nair et al., 2021; Kidambi et al., 2021; Fujimoto et al., 2019). Recent works have also

explored modern generative architectures for value function learning (Dong et al., 2025d; Agrawalla et al., 2025). Our work also proposes a novel value function design, but with an explicit focus on scalability and enabling effective performance improvements as model capacity increases. The techniques we introduce are largely orthogonal to these prior works as TQL, in principle, can be combined with these different algorithms and distribution choices.

**Transformers in RL.** Transformers have demonstrated significant success in RL in trajectory modeling (Wu et al., 2023; Zheng et al., 2022; Furuta et al., 2022; Chen et al., 2021; Janner et al., 2021), world modeling (Cheng et al., 2025), and joint policy-value architectures (Springenberg et al., 2024). In contrast, we focus on the question of how to effectively scale transformers specifically for value functions. While there have been some works that explore designing transformer-based value functions (Obando-Ceron et al., 2024; Chebotar et al., 2023), the reason transformers do not scale directly for value functions remains unclear, and we lack a reliable recipe for large-scale transformer-based value function training. Physical Intelligence et al. (2025a) explores training value functions with Monte Carlo returns, whereas we explore how to train such value functions with bootstrapping, which has been shown to work better but often has more training instabilities. Various other works have examined training instabilities that arise from scaling transformers in other domains such as NLP (Zhai et al., 2023; Wortsman et al., 2023; Zhuo et al., 2025). In contrast to these approaches, we focus exclusively on value function modeling with transformers, investigating architectural choices that enable stable and effective scaling in a value function training setting.

**Scaling Parameters in RL.** Scaling parameter sizes in RL remains challenging compared to supervised learning do-

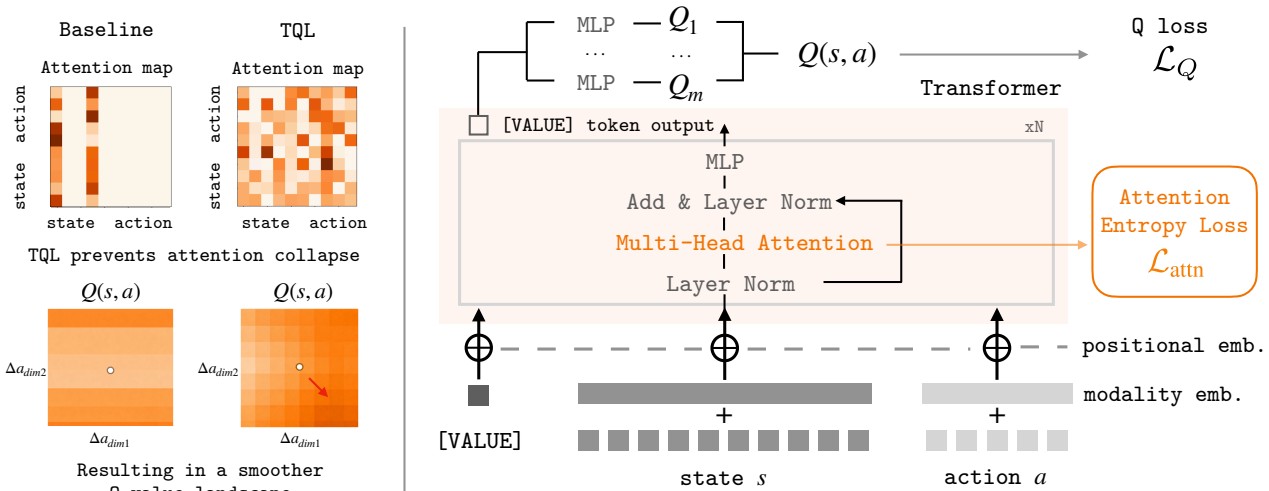

*Figure 2.* **Scaling value networks with TQL.** TQL prevents attention collapse in larger networks by controlling the target entropy of attention scores (right). Compared to an unregularized model, TQL exhibits a more uniform attention distribution and a smoother value landscape (left).

mains. Existing approaches primarily address this through regularization. One line of work employs periodic network resets to maintain plasticity during training (Nauman et al., 2024; Schwarzer et al., 2023), though this is compute-heavy as the network needs to be periodically retrained from scratch and it risks catastrophic forgetting of previously learned actions, which can make online training in the real world unsafe. Other approaches have explored architectural and objective modifications through normalization techniques to stabilize training with larger networks (Lee et al., 2025), using specific architectures such as mixture-of-experts or multi-skip residual connections (Obando-Ceron et al., 2024; Castanyer et al., 2025), or using categorical losses instead of regression for learning the value function (Farebrother et al., 2024). Recent work has examined scaling network depth for both actor and critic networks (Wang et al., 2025), demonstrating improvements in self-supervised contrastive RL settings. While these methods address specific scaling challenges, they are often limiting and require specific settings or changes to the architecture, making them unable to be applied in all cases such as in the case of pretrained models. In this work, we focus on designing a simple and general method for scaling up transformers for value function training in RL.

## 3. Preliminaries

We consider a Markov decision process (MDP) defined by a tuple $\{\mathcal{S}, \mathcal{A}, \rho, r, \gamma, T\}$, where $\mathcal{S}$ is the state space, $\mathcal{A}$ is the action space, $\rho(s)$ is the initial state distribution, $r : \mathcal{S} \times \mathcal{A} \to \mathbb{R}$ is a function defining the rewards, $\gamma \in [0, 1]$ is a discount factor, and $T(s'|s, a)$ specifies the transition probability. The goal of RL is to learn a pol-

icy that maximizes the expected sum of discounted returns $\mathbb{E}_\pi[\sum_{t=0}^{T} \gamma^t r(s_t, a_t)]$. In this paper, we consider the problem of offline RL, which additionally assumes a fixed dataset $\mathcal{D} = \{(s, a, r, s')\}$ containing pre-collected transitions. The aim of offline RL is to maximize returns by learning from the fixed dataset without environment interactions.

## 4. Transformer Q-Learning

In this section, we present the framework of TQL to enable scalable value function training with transformer architectures. We identify the key challenge: naively scaling up transformers results in a collapse of the entropy of attention weights as the network struggles to learn which tokens to attend to with its extra capacity (full analysis in Section 5.2). Our key insight is that preventing this collapse of entropy enables transformers to scale effectively for value function training, and we can do so by directly controlling the entropy of the attention scores towards a target value. We first describe our setup of training value functions with transformers (Section 4.1), then present the approach for controlling attention score entropy (Section 4.2), and finally discuss implementation details, specifically the incorporation of learnable modality embeddings to help larger models better attend to task-relevant patterns and policy extraction (§4.3). The complete TQL framework is illustrated in Figure 2 and the full algorithm is described in Algorithm 1.

### 4.1. Transformer-based Value Functions

We use a transformer architecture for learning the Q-function $Q(s, a)$ that maps state-action pairs to expected returns. We use a standard transformer decoder with full

self-attention to predict value, where we treat each dimension of the state $s \in \mathbb{R}^{n_s}$ and action $a \in \mathbb{R}^{n_a}$ as a token and project each scalar to the hidden dimension. The tokens are combined with positional encodings before being processed by a stack of transformer layers. We prepend a learnable `[VALUE]` token to the sequence, whose final representation is passed through an MLP head to produce the value prediction. We use the standard Q-learning objective

$$\mathcal{L}_Q(\phi) = \mathbb{E}[(Q_\phi(s, a) - r - \gamma Q_{\phi'}(s', a'))^2] \quad (1)$$

where $Q_{\phi'}$ is a delayed copy of $Q_\phi$. This design enables the model to learn complex dependencies between state and action tokens through self-attention mechanisms while maintaining simplicity. As we demonstrate in Section 5.2, simply scaling the parameter capacity of this standard design leads to training instabilities and actually leads to *worse* performance, rather than improved performance. In fact, we find that scaling results in a predictable decrease in performance – in one case (Figure 3), the average success rate plummets from 46% to just 6%.

### 4.2. Directly Guiding Entropy of Attention Scores

We hypothesize that the poor performance of large transformer value networks can be attributed to *attention collapse*. Specifically, our experiments in Section 5.2 show that the attention distributions of large Q-networks become increasingly concentrated on a handful of tokens. Once collapsed, the model is unable to estimate accurate values of expected discounted returns and obtain high performance.

To address the challenges of unstable attention, we can directly guide the entropy of attention distributions during training toward a target value. For a given attention layer $\ell$, let $A^\ell \in \mathbb{R}^{n \times n}$ denote the attention score matrix (after softmax) of the layer, where $n = 1 + n_s + n_a$ is the total sequence length, $n_s$ is the dimension of the state, $n_a$ is the dimension of the actions, and 1 corresponds to the `[VALUE]` token. The entropy of the attention distribution for the $i$-th query token is:

$$H_i^\ell = -\sum_{j=1}^{n} A_{ij}^\ell \log A_{ij}^\ell \quad (2)$$

Inspired by maximum entropy RL (Haarnoja et al., 2018), we introduce a learnable temperature parameter $\alpha$ and optimize it to maintain a target entropy $\bar{H}$:

$$\mathcal{L}_{\text{temp}}(\alpha) = \frac{1}{L} \sum_{\ell=1}^{L} \alpha^\ell \left( H^\ell - \bar{H} \right) \quad (3)$$

$$\mathcal{L}_{\text{attn}}(\phi) = -\frac{1}{L} \sum_{\ell=1}^{L} \alpha^\ell H^\ell \quad (4)$$

where $H^\ell = \frac{1}{n} \sum_{i=1}^{n} H_i^\ell$. In practice, we parameterize $\alpha$ as an exponential $\exp(\hat{\alpha})$ and optimize $\hat{\alpha}$. By maximizing attention entropy in the attention loss, the network is encouraged to diversify its attention to focus on all of the input tokens, while ensuring stability as the temperature term adjusts entropy toward a desired level. The temperature parameter is optimized jointly with the network parameters via gradient descent to stabilize the attention entropy:

$$\mathcal{L}_{\text{critic}}(\phi, \alpha) = \mathcal{L}_Q(\phi) + \mathcal{L}_{\text{attn}}(\phi) + \mathcal{L}_{\text{temp}}(\alpha) \quad (5)$$

We make two important design choices in how the attention entropy control is applied. First, we use *layer-wise* temperature parameters $\alpha^\ell$ rather than a single global parameter. Different layers in a transformer can learn to attend to different patterns. By enabling each layer to maintain its own target entropy, we allow the model to learn the separate attention patterns for each layer that are appropriate for value function learning. Second, we apply a separate temperature $\alpha_{[\text{VALUE}]}^\ell$ for controlling the attention entropy of the `[VALUE]` token, which aggregates information from all other tokens to produce the final Q-value. The `[VALUE]` token can contain different attention entropy patterns, as it is responsible for aggregating information from all state and action tokens to produce an accurate value estimate. These design choices allow each layer and the `[VALUE]` token to independently maintain appropriate entropy levels, leading to stable training.

The automatic entropy control mechanism allows TQL to prevent entropy collapse by directly adjusting the entropy to a desired value, as the temperature coefficient adaptively regulates the entropy in attention distributions to extract information from all tokens, while still allowing the model to learn which tokens are most relevant for value prediction. This enables larger models to utilize their extra capacity to learn the most useful signals, instead of collapse. This approach introduces an additional hyperparameter $\bar{H}$ for the target entropy values. In practice, we use a similar set of values for $\bar{H}$ across tasks and environments, and provide recommendations for how to select the value of this hyperparameter in Section C.3.

### 4.3. Implementation Details

**Learnable Modality Embedding.** Beyond stabilizing the entropy of attention scores, we seek to more effectively utilize the capacity of larger models. To this end, we introduce learnable modality embeddings $e_s$ and $e_a$ that are added to state and action tokens respectively. These modality embeddings provide a simple mechanism for the model to differentiate between state and action information. Combined with positional embeddings, this allows larger models to focus the attention on states or actions depending on what information is most relevant for Q-value prediction.

**Algorithm 1** Transformer Q-Learning (TQL)

---

**Input:** Offline dataset $\mathcal{D}$, target entropy coefficient $\bar{H}$, number of layers $L$
Initialize Q-network $Q_\phi$ with transformer architecture
Initialize target network $Q_{\phi'}$ with $\phi' \leftarrow \phi$
Initialize policy network $\pi_\omega$, $\pi_\theta^\beta$, temperature parameters $\alpha = \{\alpha^\ell, \alpha^\ell_{[\text{VALUE}]}\}_{\ell=1}^L$
**for** each training iteration **do**
    Sample batch $(s, a, r, s') \sim \mathcal{D}$
    ▷ Train Critic
    Sample next actions $a' \sim \pi_\theta(a'|s')$
    Compute TD loss $\mathcal{L}_Q$ with Equation (1)
    Compute attention entropies $\{H^\ell, H^\ell_{[\text{VALUE}]}\}_{\ell=1}^L$
    Compute entropy losses $\mathcal{L}_{\text{attn}}$ with Equation (4)
    Train critic $Q_\phi$ by minimizing $\mathcal{L}_Q + \mathcal{L}_{\text{attn}}$
    Train temperatures $\alpha$ by minimizing temperature loss $\mathcal{L}_{\text{temp}}$ with Equation (3)
    Periodically update target: $\phi' \leftarrow \tau\phi + (1 - \tau)\phi'$

    ▷ Policy Extraction
    Update one-step policy $\pi_\omega$ with $\mathcal{L}_{\text{OS}}$ in Equation (7)
    Update flow policy $\pi_\theta^\beta$ with $\mathcal{L}_{\text{BC}}$ in Equation (6)
**return** $Q_\phi, \pi_\omega$

---

**Policy Extraction.** TQL can, in principle, be applied to any value-based policy extraction scheme and has no method-specific design decisions. We choose a recent, high-performing offline RL policy extraction scheme (Park et al., 2025b): we learn a one-step flow policy $\pi_\omega$ that is behavior constrained through distillation toward a Behavior Cloning (BC) flow policy $\pi^\beta$ while steering it to maximize Q-values:

$$\mathcal{L}_{\text{BC}}(\theta) = \mathbb{E}_{\substack{s, a=x^1 \sim \mathcal{D}, \\ x^0 \sim \mathcal{N}(0, I_d), \\ t \sim \text{Unif}([0,1])}} \left[ \|\pi_\theta^\beta(t, s, x^t) - (x^1 - x^0)\|_2^2 \right], \quad (6)$$

$$\mathcal{L}_{\text{OS}}(\omega) = \underbrace{\mathbb{E}_{s \sim \mathcal{D}, a^\pi \sim \pi_\omega}[-Q_\phi(s, a^\pi)]}_{Q} + \underbrace{\alpha\mathcal{L}_{\text{Distill}}(\omega)}_{\text{BC}}, \quad (7)$$

$$\mathcal{L}_{\text{Distill}}(\omega) = \mathbb{E}_{\substack{s \sim \mathcal{D}, \\ z \sim \mathcal{N}(0, I_d)}} \left[ \|\mu_\omega(s, z) - \mu_\theta(s, z)\|_2^2 \right]. \quad (8)$$

where $\alpha$ controls the strength of the behavior constraint on the policy. This allows the policy to be behavior-constrained while still maximizing value.

# 5. Experiments

In this section, we empirically evaluate TQL compared to prior RL methods on a variety of challenging tasks, focusing on scaling and performance. We start by presenting the experiment setup followed by the results. The goal of our experiments is to answer the following key questions:

**(Q1)** What prevents transformers from scaling well for value functions?

**(Q2)** Does TQL improve transformer scalability in a value function setting compared to prior approaches?

**(Q3)** Can TQL effectively learn a policy from a static offline dataset?

**(Q4)** What are the most important components of TQL?

## 5.1. Experiment Setup

We evaluate TQL on the recently proposed OGBench benchmark (Park et al., 2025a). Success on OGBench requires offline RL algorithms to reason over extended temporal horizons, learn effectively from unstructured data, and handle delayed credit assignment across complex, multi-stage interactions, making it a rigorous testbed for evaluating offline RL algorithms. We adopt the reward-based single-task variants (denoted as "singletask") as we are in the setting of standard reward-maximizing RL. Our evaluation spans five domains of five distinct tasks each: `cube-double`, `cube-triple`, `scene`, `puzzle-3×3`, and `puzzle-4×4`, resulting in a total of 25 tasks. We illustrate the domains in Figure 14.

**Comparisons.** In our experiments, we compare against 9 offline RL methods selected to represent a broad range of algorithms and modeling strategies, including methods that use flow-matching-based and transformer-based value functions. For standard offline RL comparisons using Gaussian policies, we include Behavior Cloning (BC), Implicit Q Learning (IQL) (Kostrikov et al., 2021), and ReBRAC (Tarasov et al., 2023) as widely adopted and competitive representatives. To capture more recent state-of-the-art advances, we further evaluate methods with flow policies in offline RL, including FBRAC which is a variant of Behavior Regularized Actor Critic (Wu et al., 2019) with flow policies and IFQL, which is a variant of Implicit Diffusion Q-Learning with flow policies (Hansen-Estruch et al., 2023), and Flow Q-Learning (FQL) (Park et al., 2025b) which uses a one-step flow policy to maximize the Q-value. Because our method focuses on value learning, we include baselines with flow-based value learning (floq (Agrawalla et al., 2025)) and transformer-based value learning (Q-Transformer (Q-T) (Chebotar et al., 2023) and Perceiver Actor Critic (PAC) (Springenberg et al., 2024)). For both PAC and Q-T, we follow their original value function architectures and training setups, while adopting the same FQL based policy extraction procedure as our method to ensure a fair and controlled comparison. For detailed baseline implementations and hyperparameters, we provide them in Appendix C.1.

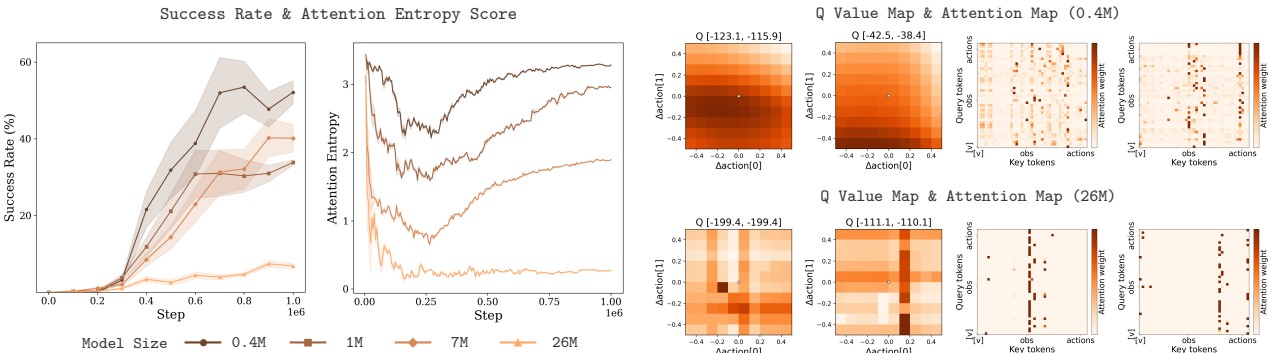

*Figure 3.* **Scaling transformers for value functions results in entropy collapse and worse performance.** Left: Visualizations of the success rate and attention entropy of the transformer model without TQL across five `cube-double` tasks under different model sizes. Right: Q-value landscapes and attention maps for the smallest (0.4M) and largest (26M) models. The larger transformer learns highly non-smooth value surfaces and exhibits high-frequency oscillations and discontinuities that are absent in its smaller counterpart.

| | MLP Q | | | | | | Flow Q | Transformer Q | | |
|---|---|---|---|---|---|---|---|---|---|---|
| | BC | IQL | ReBRAC | FBRAC | IFQL | FQL | floq | Q-T | PAC | TQL |
| `cube-double-play-singletask-task1-v0` | $8 \pm 3$ | $27 \pm 5$ | $45 \pm 6$ | $47 \pm 11$ | $35 \pm 9$ | $61 \pm 9$ | $45 \pm 23$ | $2 \pm 1$ | $28 \pm 16$ | $\mathbf{95 \pm 2}$ |
| `cube-double-play-singletask-task2-v0` | $0 \pm 0$ | $1 \pm 1$ | $7 \pm 3$ | $22 \pm 12$ | $9 \pm 5$ | $36 \pm 6$ | $42 \pm 23$ | $0 \pm 0$ | $36 \pm 16$ | $\mathbf{84 \pm 8}$ |
| `cube-double-play-singletask-task3-v0` | $0 \pm 0$ | $0 \pm 0$ | $4 \pm 1$ | $4 \pm 2$ | $8 \pm 5$ | $22 \pm 5$ | $\mathbf{59 \pm 15}$ | $0 \pm 0$ | $50 \pm 11$ | $60 \pm 27$ |
| `cube-double-play-singletask-task4-v0` | $0 \pm 0$ | $0 \pm 0$ | $1 \pm 1$ | $0 \pm 1$ | $1 \pm 1$ | $5 \pm 2$ | $13 \pm 8$ | $0 \pm 0$ | $10 \pm 6$ | $\mathbf{14 \pm 6}$ |
| `cube-double-play-singletask-task5-v0` | $0 \pm 0$ | $4 \pm 3$ | $4 \pm 2$ | $2 \pm 2$ | $17 \pm 6$ | $19 \pm 10$ | $46 \pm 11$ | $0 \pm 0$ | $25 \pm 21$ | $\mathbf{84 \pm 9}$ |
| `cube-double-play (average)` | $2 \pm 1$ | $6 \pm 2$ | $12 \pm 3$ | $15 \pm 6$ | $14 \pm 5$ | $29 \pm 6$ | $41 \pm 16$ | $0 \pm 0$ | $30 \pm 14$ | $\mathbf{67 \pm 10}$ |
| `scene-play-singletask-task1-v0` | $19 \pm 6$ | $94 \pm 3$ | $\mathbf{95 \pm 2}$ | $\mathbf{96 \pm 8}$ | $\mathbf{98 \pm 3}$ | $\mathbf{100 \pm 0}$ | $\mathbf{100 \pm 0}$ | $27 \pm 6$ | $\mathbf{100 \pm 0}$ | $\mathbf{100 \pm 0}$ |
| `scene-play-singletask-task2-v0` | $1 \pm 1$ | $12 \pm 3$ | $50 \pm 13$ | $46 \pm 10$ | $0 \pm 0$ | $76 \pm 9$ | $\mathbf{92 \pm 9}$ | $6 \pm 3$ | $54 \pm 16$ | $81 \pm 14$ |
| `scene-play-singletask-task3-v0` | $1 \pm 1$ | $32 \pm 7$ | $55 \pm 16$ | $78 \pm 4$ | $54 \pm 19$ | $\mathbf{98 \pm 1}$ | $97 \pm 4$ | $1 \pm 1$ | $\mathbf{98 \pm 2}$ | $\mathbf{98 \pm 2}$ |
| `scene-play-singletask-task4-v0` | $2 \pm 2$ | $0 \pm 1$ | $3 \pm 3$ | $4 \pm 4$ | $0 \pm 0$ | $5 \pm 1$ | $4 \pm 4$ | $1 \pm 1$ | $4 \pm 3$ | $\mathbf{25 \pm 17}$ |
| `scene-play-singletask-task5-v0` | $\mathbf{0 \pm 0}$ | $\mathbf{0 \pm 0}$ | $\mathbf{0 \pm 0}$ | $\mathbf{0 \pm 0}$ | $\mathbf{0 \pm 0}$ | $\mathbf{0 \pm 0}$ | $\mathbf{0 \pm 0}$ | $\mathbf{0 \pm 1}$ | $\mathbf{0 \pm 0}$ | $\mathbf{0 \pm 0}$ |
| `scene-play (average)` | $5 \pm 2$ | $28 \pm 3$ | $41 \pm 7$ | $45 \pm 5$ | $30 \pm 4$ | $56 \pm 2$ | $\mathbf{59 \pm 3}$ | $7 \pm 2$ | $51 \pm 4$ | $\mathbf{61 \pm 7}$ |
| `puzzle-3x3-play-singletask-task1-v0` | $5 \pm 2$ | $33 \pm 6$ | $\mathbf{97 \pm 4}$ | $63 \pm 19$ | $\mathbf{94 \pm 3}$ | $90 \pm 4$ | $\mathbf{94 \pm 5}$ | $3 \pm 2$ | $92 \pm 4$ | $\mathbf{96 \pm 4}$ |
| `puzzle-3x3-play-singletask-task2-v0` | $1 \pm 1$ | $4 \pm 3$ | $1 \pm 1$ | $2 \pm 2$ | $1 \pm 2$ | $16 \pm 5$ | $\mathbf{22 \pm 9}$ | $1 \pm 1$ | $17 \pm 12$ | $19 \pm 12$ |
| `puzzle-3x3-play-singletask-task3-v0` | $1 \pm 1$ | $3 \pm 2$ | $3 \pm 1$ | $1 \pm 1$ | $0 \pm 0$ | $10 \pm 3$ | $\mathbf{16 \pm 5}$ | $0 \pm 1$ | $8 \pm 4$ | $7 \pm 5$ |
| `puzzle-3x3-play-singletask-task4-v0` | $1 \pm 1$ | $2 \pm 1$ | $2 \pm 1$ | $2 \pm 2$ | $0 \pm 0$ | $\mathbf{16 \pm 5}$ | $\mathbf{16 \pm 10}$ | $0 \pm 0$ | $14 \pm 3$ | $10 \pm 4$ |
| `puzzle-3x3-play-singletask-task5-v0` | $1 \pm 0$ | $3 \pm 2$ | $5 \pm 3$ | $2 \pm 2$ | $0 \pm 0$ | $16 \pm 3$ | $25 \pm 7$ | $0 \pm 1$ | $9 \pm 3$ | $\mathbf{30 \pm 10}$ |
| `puzzle-3x3-play (average)` | $2 \pm 1$ | $9 \pm 3$ | $22 \pm 2$ | $14 \pm 5$ | $19 \pm 1$ | $30 \pm 4$ | $\mathbf{35 \pm 7}$ | $1 \pm 1$ | $28 \pm 5$ | $32 \pm 7$ |
| `puzzle-4x4-play-singletask-task1-v0` | $1 \pm 1$ | $12 \pm 2$ | $26 \pm 4$ | $32 \pm 9$ | $49 \pm 9$ | $34 \pm 8$ | $\mathbf{60 \pm 8}$ | $0 \pm 0$ | $14 \pm 7$ | $\mathbf{59 \pm 11}$ |
| `puzzle-4x4-play-singletask-task2-v0` | $0 \pm 0$ | $7 \pm 4$ | $12 \pm 4$ | $5 \pm 3$ | $4 \pm 4$ | $16 \pm 5$ | $\mathbf{22 \pm 5}$ | $0 \pm 1$ | $13 \pm 5$ | $19 \pm 6$ |
| `puzzle-4x4-play-singletask-task3-v0` | $0 \pm 0$ | $9 \pm 3$ | $15 \pm 3$ | $20 \pm 10$ | $\mathbf{50 \pm 14}$ | $18 \pm 5$ | $36 \pm 7$ | $0 \pm 0$ | $8 \pm 7$ | $43 \pm 5$ |
| `puzzle-4x4-play-singletask-task4-v0` | $0 \pm 0$ | $5 \pm 2$ | $10 \pm 3$ | $5 \pm 1$ | $\mathbf{21 \pm 11}$ | $11 \pm 3$ | $19 \pm 4$ | $0 \pm 0$ | $3 \pm 2$ | $20 \pm 3$ |
| `puzzle-4x4-play-singletask-task5-v0` | $0 \pm 0$ | $4 \pm 1$ | $7 \pm 3$ | $2 \pm 2$ | $2 \pm 2$ | $7 \pm 3$ | $14 \pm 3$ | $0 \pm 0$ | $5 \pm 3$ | $\mathbf{16 \pm 5}$ |
| `puzzle-4x4-play (average)` | $0 \pm 0$ | $7 \pm 2$ | $14 \pm 3$ | $13 \pm 5$ | $25 \pm 8$ | $17 \pm 5$ | $\mathbf{30 \pm 5}$ | $0 \pm 0$ | $9 \pm 5$ | $31 \pm 6$ |
| `cube-triple-play-singletask-task1-v0` | $1 \pm 1$ | $4 \pm 4$ | $1 \pm 2$ | $0 \pm 0$ | $2 \pm 2$ | $20 \pm 6$ | $17 \pm 11$ | $0 \pm 0$ | $1 \pm 1$ | $\mathbf{29 \pm 16}$ |
| `cube-triple-play-singletask-task2-v0` | $0 \pm 0$ | $0 \pm 0$ | $0 \pm 0$ | $0 \pm 0$ | $0 \pm 0$ | $1 \pm 2$ | $0 \pm 0$ | $0 \pm 0$ | $0 \pm 0$ | $0 \pm 1$ |
| `cube-triple-play-singletask-task3-v0` | $0 \pm 0$ | $0 \pm 0$ | $0 \pm 0$ | $0 \pm 0$ | $0 \pm 0$ | $0 \pm 0$ | $1 \pm 1$ | $0 \pm 0$ | $0 \pm 0$ | $\mathbf{5 \pm 4}$ |
| `cube-triple-play-singletask-task4-v0` | $\mathbf{0 \pm 0}$ | $\mathbf{0 \pm 0}$ | $\mathbf{0 \pm 0}$ | $\mathbf{0 \pm 0}$ | $\mathbf{0 \pm 0}$ | $\mathbf{0 \pm 0}$ | $\mathbf{0 \pm 0}$ | $\mathbf{0 \pm 0}$ | $\mathbf{0 \pm 0}$ | $\mathbf{0 \pm 0}$ |
| `cube-triple-play-singletask-task5-v0` | $0 \pm 0$ | $\mathbf{1 \pm 1}$ | $0 \pm 0$ | $0 \pm 0$ | $0 \pm 0$ | $0 \pm 0$ | $0 \pm 0$ | $0 \pm 0$ | $0 \pm 0$ | $0 \pm 0$ |
| `cube-triple-play (average)` | $0 \pm 0$ | $1 \pm 1$ | $0 \pm 0$ | $0 \pm 0$ | $0 \pm 0$ | $4 \pm 2$ | $4 \pm 2$ | $0 \pm 0$ | $0 \pm 0$ | $\mathbf{7 \pm 4}$ |
| **Average across all tasks** | $2 \pm 1$ | $10 \pm 2$ | $18 \pm 3$ | $17 \pm 4$ | $18 \pm 4$ | $27 \pm 4$ | $34 \pm 7$ | $2 \pm 1$ | $24 \pm 6$ | $\mathbf{40 \pm 7}$ |

*Table 1.* **OGBench evaluation results.** TQL achieves the highest average performance on 4 out of 5 domains, as well as the best average performance across all 25 tasks. The reported scores are averaged over 3 seeds, where bold values indicate performance within 95% of the best result per task.

## 5.2. What Prevents Transformers from Scaling Effectively for Value Functions?

While transformers have demonstrated scalability across numerous domains, their application to value function learning has been limited. We first investigate the specific bot-

tlenecks preventing effective scaling of transformer-based value functions through an empirical analysis. We conduct scaling experiments on the `cube-double` domain consisting of 5 tasks using the transformer architecture described in Section 4.1, varying model capacity while maintaining the same architecture. The training curves for different parame-

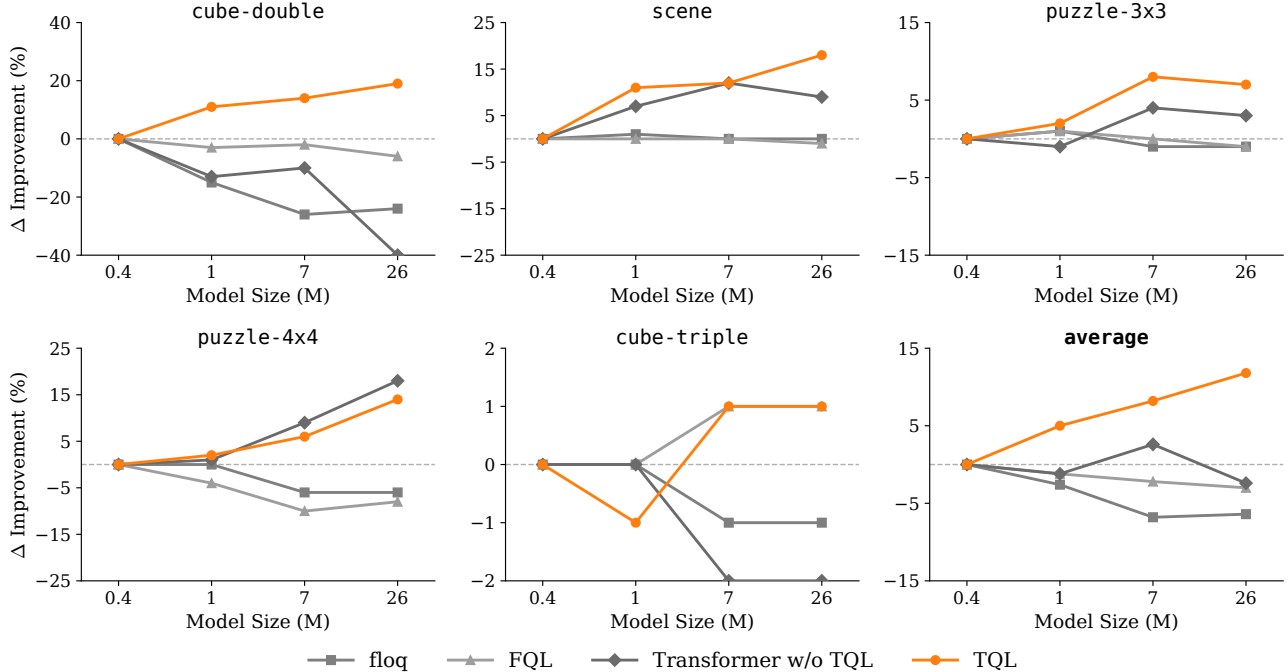

*Figure 4.* **Scaling results.** We compare TQL against prior offline RL methods across critic sizes from 0.4M to 26M parameters. The plot reports the average success rate difference compared to the smallest model (0.4M) for each method. While baselines suffer from performance degradation at larger scales, TQL consistently scales well across all environments, outperforming prior methods by a large margin.

ter counts are presented in Figure 3 (left). Contrary to the typical scaling trends observed in language modeling and vision tasks, we observe a strong negative scaling pattern: performance degrades with increased model size, with the largest model performing poorly.

To diagnose this failure mode, we analyze the Q-value landscapes using a contour visualization in Figure 3 (center right). We observe that larger networks produce increasingly non-smooth value surfaces and exhibit high-frequency oscillations and discontinuities that are absent in their smaller counterparts. This suggests an instability that occurs with larger networks and that excess capacity appears detrimental. We hypothesize that this degradation is a result of training instabilities in the attention mechanism in transformers. To test this hypothesis, we examine the attention score distributions across network scales in Figure 3 (right). Consistent with the hypothesis, we find that attention entropy decreases substantially with model size, indicating that larger models learn increasingly peaked and brittle attention patterns. This entropy collapse correlates strongly with both the non-smooth value landscapes and diminished task performance, suggesting that the attention mechanism fails to generalize appropriately when scaled without modifications in the setting of value function learning. We refer to Section A.1 and Section A.2 for additional attention entropy plots and visualizations for transformer without TQL across different environments, which exhibit consistent trends.

### 5.3. Scaling TQL with Parameter Sizes

Building upon the empirical analysis in Section 5.2, we conduct scaling experiments to examine whether TQL mitigates attention entropy collapse and enables effective scaling of Q-functions. We compare TQL with representative methods for learning the Q-function in the offline RL setting across a variety of generative model backbones, including FQL (Park et al., 2025b) (MLP), floq (Agrawalla et al., 2025) (flow-matching), and PAC (Springenberg et al., 2024) (transformer). We additionally include a transformer baseline as described in Section 4.1, which has the same architecture as TQL but without the attention entropy regularization and modality tokens. All methods are evaluated across a range of critic network sizes ranging from 0.4M to 26M parameters (0.4M, 1M, 7M, and 26M). We note that traditionally, RL methods use network sizes of ~1M parameters to train the Q-function, as evidenced in prior works (Dong et al., 2025d; Agrawalla et al., 2025; Kostrikov et al., 2021; Fujimoto & Gu, 2021; Kumar et al., 2020) and also by which sizes give the best performance for baselines in our scaling results.

We present the results in Figure 4. From the plots, we see that across all generative model backbones (MLP, flow-matching, transformer), prior methods scale poorly with additional capacity, with an average decrease in performance of 10.6% from the smallest to largest settings. This suggests that existing offline RL methods, which may have strong performance on offline RL benchmarks, all struggle to ben-

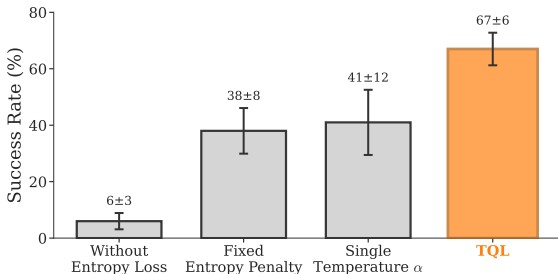

*Figure 5.* **Ablation on the most important components of TQL.** We compare TQL against four ablated variants on the `cube-double` environment: (1) a transformer baseline without attention entropy guidance, (2) using a fixed entropy penalty, and (3) using one temperature instead of layer-wise and token-wise temperatures. The results represent the average performance and standard error across 5 tasks and 3 seeds in the `cube-double`.

efit from larger value function capacity. Comparing to the transformer baseline, while the performance of the baseline improves on some tasks when scaled up to 7M parameters, it deteriorates at 26M. This behavior is consistent with the attention collapse patterns, as the 26M parameter model shows the most severe collapse and highly non-smooth Q-value landscapes. In contrast to prior methods, TQL mitigates this failure mode and achieves stable and consistent scaling, with a 43% performance improvement from the smallest to the largest model, highlighting the ability of TQL to effectively leverage larger capacity for improvement in performance.

### 5.4. Comparing to Prior Offline RL Methods

Having demonstrated scaling of TQL, we further evaluate on a set of benchmark tasks comparing to prior methods to see how well TQL can be used to learn a policy from offline datasets. We compare TQL against a comprehensive set of offline RL baselines, reporting numbers from the original paper for methods that test on the same benchmarks, or from tuned hyperparameter settings for those that do not. We refer to Section C.4 for more details. As summarized in Table 1, TQL achieves the best performance on 4 out of 5 domains, as well as the best average performance across all 25 tasks, demonstrating consistent improvements across a wide range of environments. These results highlight the effectiveness of TQL in achieving strong performance in challenging tasks by preventing attention collapse of transformers, resulting in a method that both scales and achieves state-of-the-art performance.

### 5.5. Which Components of TQL are Most Important?

Having established both scaling and performance, we analyze the key components of TQL and how they affect overall performance. We consider the following settings: (1) no entropy loss, which removes the loss term $\mathcal{L}_{\text{attn}}$ in Equation (5); (2) fixed entropy penalty, which replaces the entropy adjustment mechanism with a max entropy loss to

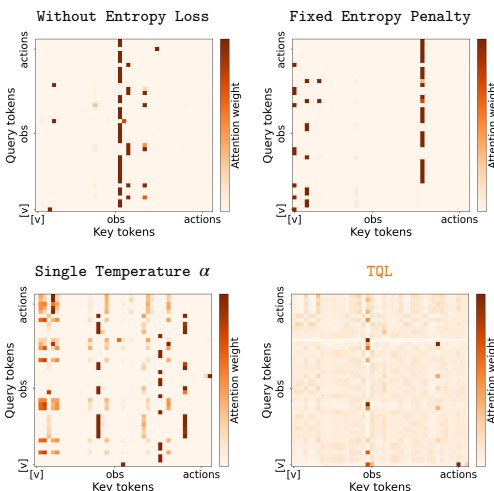

*Figure 6.* **Attention maps of ablations.** We find that TQL is the most effective at preventing attention entropy collapse, while both using a fixed entropy penalty and a single temperature across layers and tokens resulted in some degree of over-specialized attention and worse performance.

prevent attention collapse; (3) single temperature $\alpha$, which uses a single temperature across layers and tokens. For each variant, we report performance averaged across 5 tasks in the `cube-double` environment and using the 26M model.

The results are shown in Figure 5. First, we find that removing entropy guidance entirely leads to a substantial performance drop from attention collapse as we showed in Section 5.2. All forms of entropy adjustment significantly improve performance by preventing attention collapse. We find that using a fixed entropy penalty through a max entropy loss instead of entropy guidance toward a target value results in significantly worse performance, likely because of the instability in the training process arising from the max entropy loss without guiding the entropy toward a target. In addition, layer-wise and token-wise entropy tuning is also important for more stable training, as it is difficult for one temperature term to guide the entropy of all layers toward the same target. We show the attention maps of ablations in Figure 6. From the attention maps, we see that TQL has the most balanced attention across all state and action tokens, which that results in the most effective scaling and highest performance.

## 6. Discussion and Limitations

We introduce TQL, a method for controlling attention entropy which allows for the scaling of transformer-based value functions without collapse. Our work identified that attention entropy collapse is common in transformer-based value functions and found that this collapse has a strong negative correlation with task performance.

While our method demonstrates effective scaling and strong performance on benchmarks, it does have limitations. In

terms of implementation complexity, our method adds an additional loss term, as well as an extra hyperparameter for the target entropy. While this adds some complexity and tuning, we found the same set of hyperparameters to work well in our experiments as discussed in Section C.3 and, unlike other training stability tricks, our method does not require any architecture changes, enabling the use of pretrained backbones without costly retraining.

## Acknowledgments

This work is in part supported by the RAI Institute, ONR grant N00014-22-1-2293, ONR grant N00014-22-1-2621, NSF #1941722, and the NSF CAREER award.

## Impact Statement

This paper presents work whose goal is to advance the field of machine learning, specifically the scalability of reinforcement learning by enabling the use of larger transformer-based value functions. There are many potential societal consequences of our work, including applications to robotics and decision-making systems, where more scalable value functions may lead to better performance and generalization. However, there are no particular societal consequences that we feel must be specifically highlighted here.

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

# A. Full Scaling Analysis

## A.1. Full Attention Entropy Plots for Transformer Without TQL

In this section, we examine the attention entropy of a Transformer without TQL across all five environments, as shown in Figure 7. We observe a consistent trend where attention entropy decreases as model size increases, eventually collapsing to near-zero values for the largest models. This empirical evidence validates the attention collapse problem identified in our analysis.

Furthermore, we find that these entropy profiles are highly predictive of the scaling behavior reported in Figure 4. Specifically, the `puzzle-3x3` and `puzzle-4x4` environments exhibit the least severe entropy collapse in Figure 7. Correspondingly, the Transformer without TQL is able to achieve slight positive scaling on these same tasks in Figure 4. This strong correlation between sustained attention entropy and scaling ability reinforces our hypothesis that attention collapse is a primary bottleneck for performance at scale.

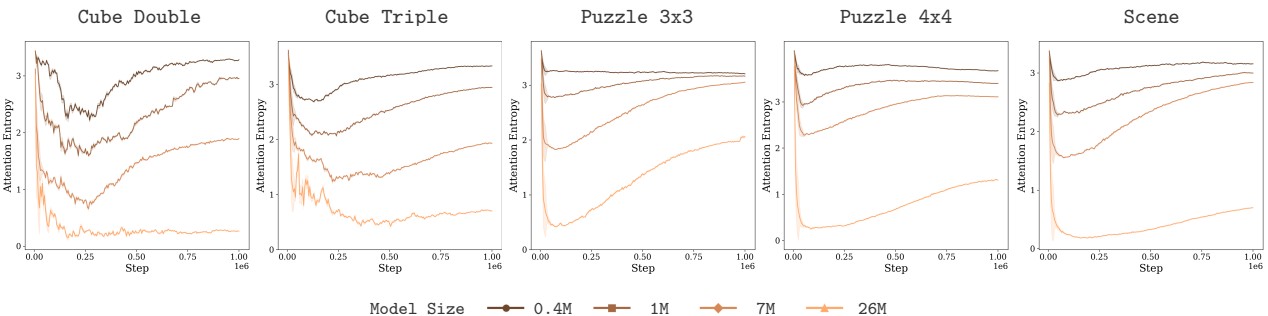

*Figure 7.* **Attention entropy analysis for Transformer without TQL.** We visualize attention entropy across 5 environments as model size increases. The plots demonstrate a consistent trend of decreasing entropy with larger models, providing empirical evidence of the attention collapse phenomenon.

## A.2. Attention Map and Q-Value Visualization

In this section, we present comprehensive visualizations comparing the attention maps and Q-value landscapes. We evaluate TQL with 26M parameters against a Transformer without TQL baseline trained without attention entropy tuning. To ensure a fair comparison, we utilize identical state and action pairs drawn from random successful trajectories across five environments.

Figure 8 displays the results for the Transformer without TQL. We observe that the attention patterns become extremely sparse and concentrate heavily on a distinct few tokens. This collapse leads to Q-value maps that are highly discontinuous and irregular. Conversely, Figure 9 demonstrates that our training objective successfully mitigates this issue. The attention mechanism maintains a broader distribution, resulting in a significantly smoother and more consistent Q-value landscape.

Large Transformer Value Function without TQL

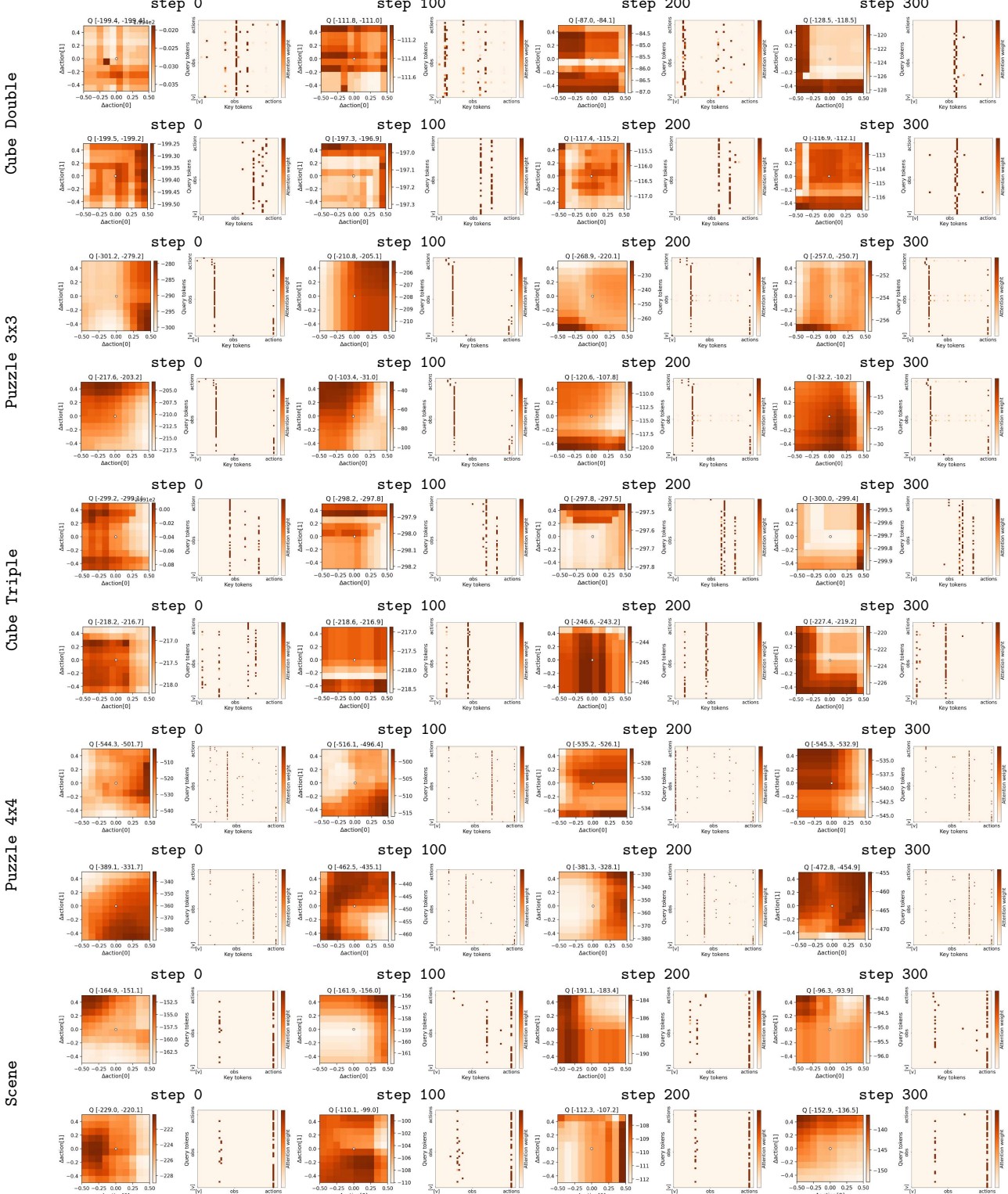

*Figure 8.* Attention maps and Q-value landscapes of a Transformer without TQL across all five environments. The learned Q-value functions exhibit non-smooth landscapes, accompanied by overfitted attention patterns, which hinder effective value learning.

Large Transformer Value Function with TQL

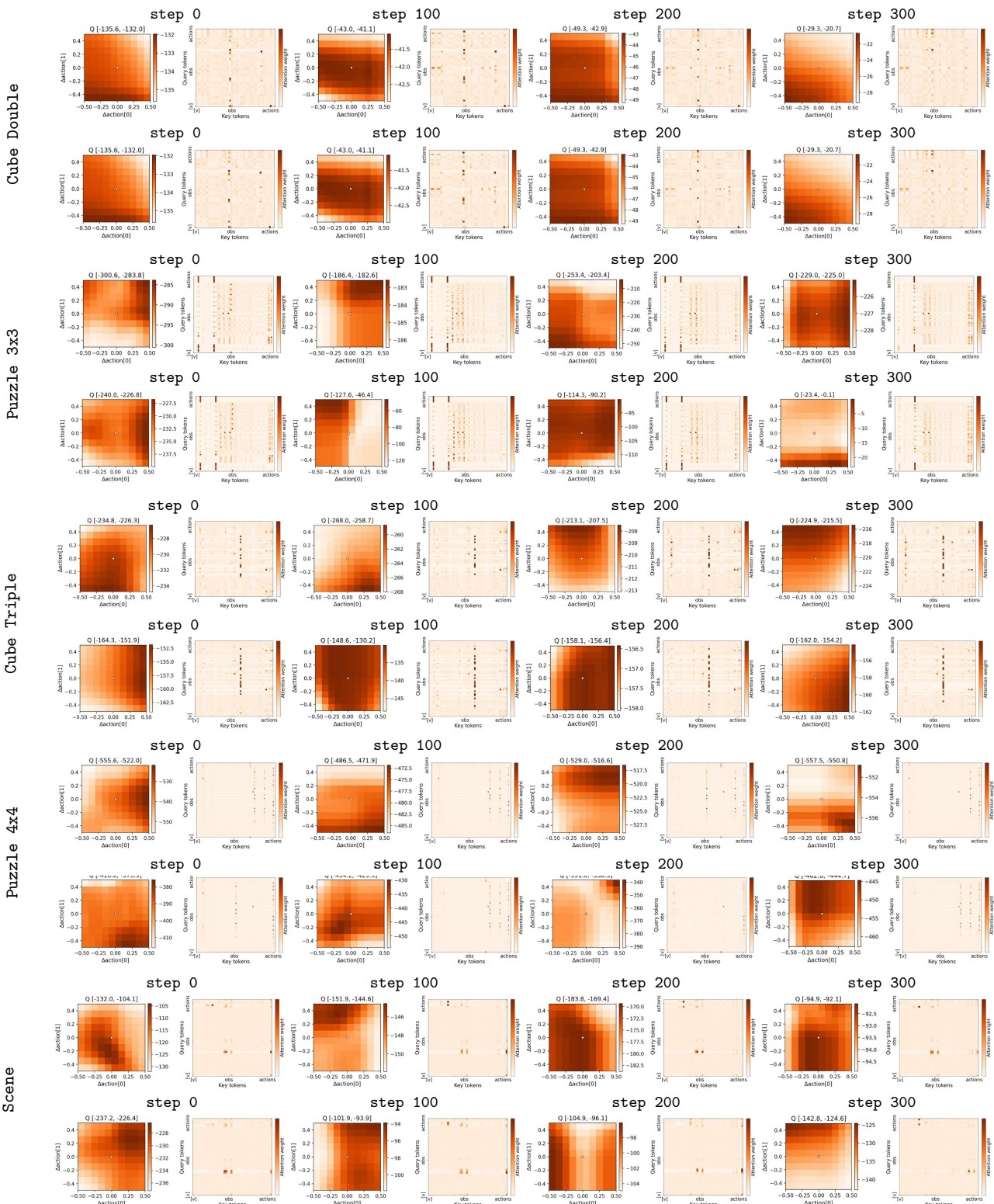

*Figure 9.* Attention maps and Q-value maps of TQL across all five environments. The learned Q-value functions exhibit smooth landscapes, accompanied by more distributed attention patterns, which facilitate stable and effective value learning.

# B. Additional Experiments

## B.1. Increasing Depth

In our primary experiments, we scale the transformer network by increasing the hidden dimension. In this section, we show another common kind of scaling for the model described in Section 4.1 as well as TQL: increasing the depth of the network. To do so, we run a parameter-controlled experiment, fixing the total number of parameters at our largest setting of around 26M. We scale to a depth of 4 layers and pick the hidden dimension such that the total parameters match the 26M-parameter setting as closely as possible. As in our primary scaling experiments in Section 5.3, we follow standard practice in keeping a fixed hidden dimension per attention head, in our case 32. We present the results in Figure 10, attention maps in Figure 11. As we see from the results, increasing depth for the baseline transformer results in the same entropy collapse and poor performance as in the case of increasing hidden dimensions, which is expected as increasing depth also increases capacity similar to increasing width. As in the width setting, TQL addresses this by preventing entropy collapse and resulting in increased performance. In fact, TQL achieves very similar performance with 2 layers and 4 layers (67% vs. 66%), suggesting that total network size is a good indicator of performance.

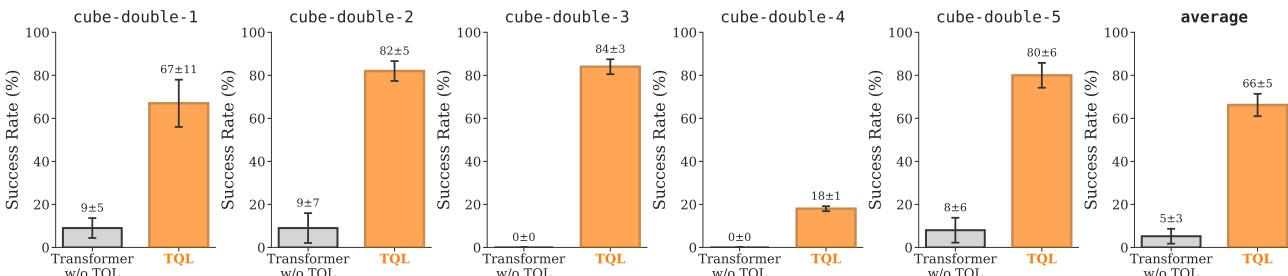

*Figure 10.* **Performance of TQL with more layers.** A transformer without TQL results in the same attention collapse as in the setting with fewer layers and TQL addresses this to improve performance. We find that the increased depth results in similar performance in both cases.

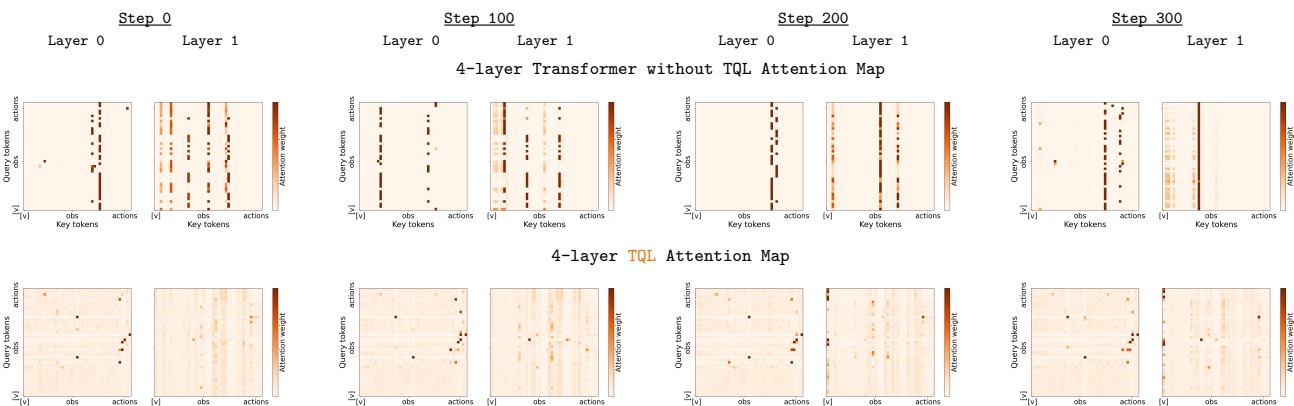

*Figure 11.* **Attention Map of TQL with increased depth.** We find that the attention collapse without TQL and the more uniform attention scores with TQL are consistent, regardless of depth.

## B.2. Comparing to Approaches for Stabilizing Transformers in Supervised Learning

In supervised learning, numerous approaches have been proposed to improve the stability of training large-scale transformers. These approaches are typically intended to allow practitioners to push the stability boundary of training, for example to train at higher learning rates (Rybakov et al., 2024), whereas in stable regimes, these methods are largely ineffective.

In this section, we analyze the performance of these approaches for preventing entropy collapse—and the subsequent performance drop—in the setting of training value functions, focusing on three primary approaches:

- QK Normalization (Dehghani et al., 2023), a method which adds a LayerNorm after both the query and key projections

- $\sigma$Reparam (Zhai et al., 2023), a method based on spectral normalization (Yoshida & Miyato, 2017) proposed to stabilize attention entropy

- A transformer implementation with a set of techniques used to stably train larger models which contains RMS Norm (Zhang & Sennrich, 2019), Sandwich Norm (Ding et al., 2021), QK Norm (Dehghani et al., 2023), and SwiGLU (Shazeer, 2020)

We present the performance in Figure 12 and the corresponding attention entropy in Figure 13. From the figure, we see that these approaches can help mitigate entropy collapse and result in higher entropy values and performance compared to the transformer baseline. However, these approaches are not able to consistently do so and perform worse in general compared to TQL when applied to value learning.

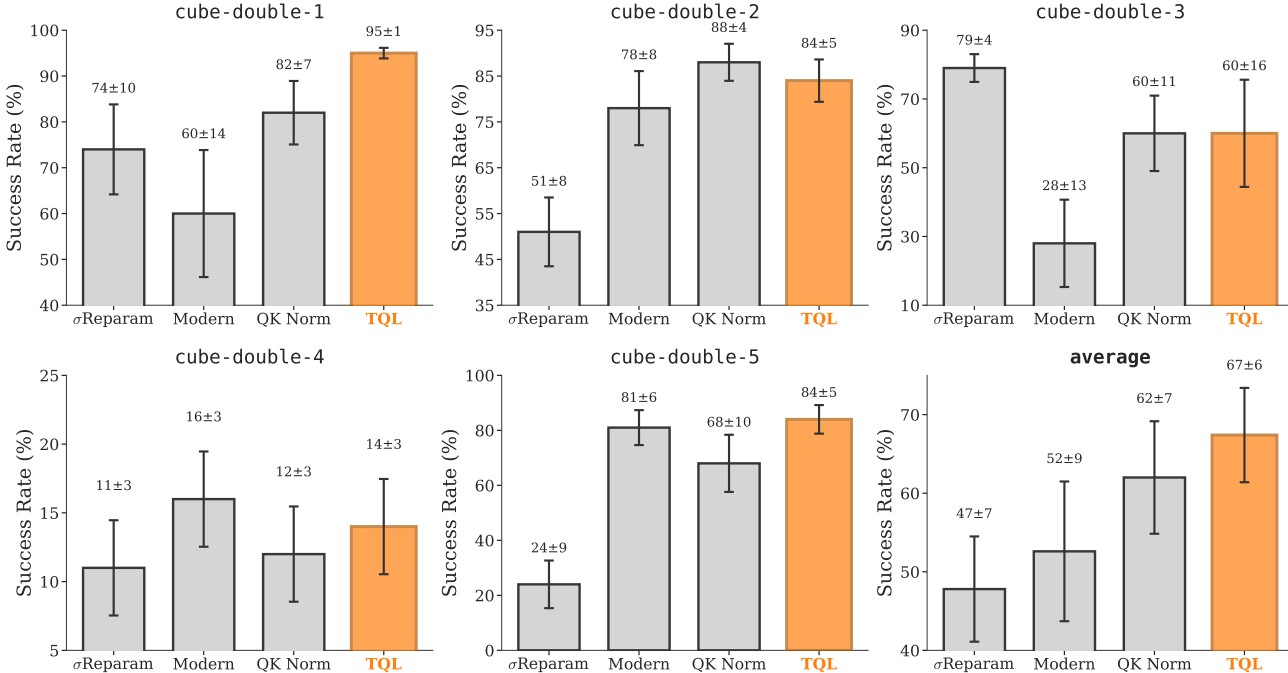

*Figure 12.* **Performance of other transformer stabilization techniques compared with TQL.** TQL achieves higher performance compared to approaches to stabilize transformer training in supervised learning.

## C. Experimental Details

### C.1. Benchmark & Evaluation

We evaluate our method on the OGBench benchmark suite (Park et al., 2025a). OGBench is a large-scale benchmark designed for offline goal-conditioned reinforcement learning and additionally provides single-task variants that are compatible with standard reward-maximizing RL algorithms. In this work, we adopt the single-task variants for all domains. For each task, the reward ranges from $-n_{\text{task}}$ to $0$, depending on the number of completed subtasks.

We use the following OGBench datasets for each domain:

- `cube-double-play-v0`

- `cube-triple-play-v0`

- `scene-play-v0`

- `puzzle-3x3-play-v0`

- `puzzle-4x4-play-v0`

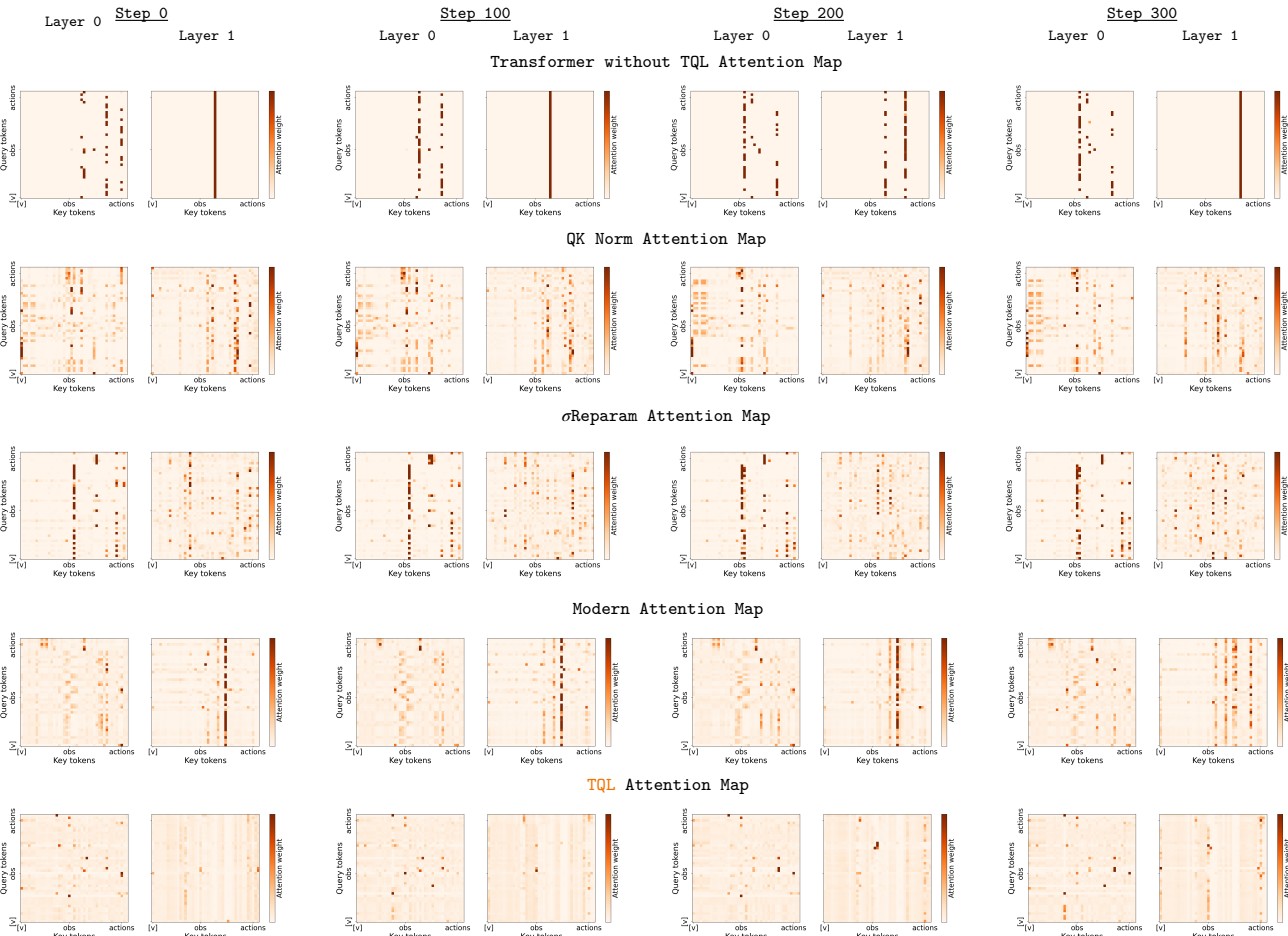

*Figure 13.* **Attention Map of other transformer stabilization techniques compared with TQL.** We see that common approaches for stabilizing transformers in supervised learning indeed result in higher entropy compared to the vanilla transformer. However, we see that for some layers the entropy can still collapse, whereas TQL has a more uniform attention pattern.

The `cube-double` and `cube-triple` tasks involve complex pick-and-place manipulation of multiple colored cube blocks. The `scene` tasks require long-horizon reasoning and interaction with diverse objects in cluttered environments. The `puzzle-3x3` and `puzzle-4x4` tasks are based on the *Lights Out* puzzle and are solved using a robotic arm, further evaluating the agent's ability to generalize over combinatorial state spaces. Visualizations of all environments are shown in Figure 14.

All experiments follow the official evaluation protocols and metrics defined by OGBench (Park et al., 2025a). For each setting, we run three random seeds and report the mean and standard deviation of the success rate, averaged over the 800k, 900k, and 1M training step checkpoints. For bar plots, we report the same average with standard error as error bars.

## C.2. Model Structure Details

We parameterize the Q-function $Q(s, a)$ with a Transformer that models interactions between state and action components at the token level. Given a state $s \in \mathbb{R}^{n_s}$ and an action $a \in \mathbb{R}^{n_a}$, we construct a token sequence by treating each scalar dimension of the state and action as an individual token. Specifically, $s$ and $a$ are reshaped into sequences of length $n_s$ and $n_a$, respectively, and each token is independently projected into a shared hidden space of dimension $H$ using learned linear projections. To distinguish state tokens from action tokens, we add learnable modality embeddings to the corresponding token representations. The state and action token sequences are concatenated to form a sequence of length $n_s + n_a$.

Learnable positional embeddings are added to all tokens, and a [value] token is prepended to the sequence to aggregate

cube-double      scene      puzzle-3x3      puzzle-4x4      cube-triple

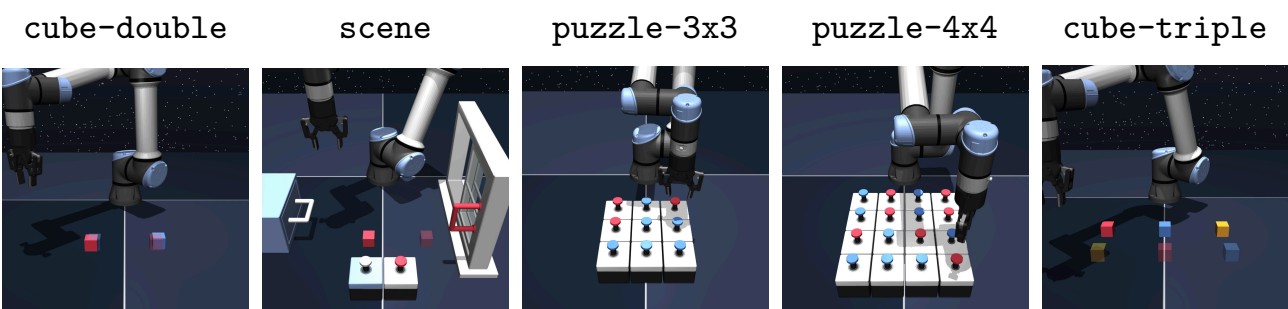

*Figure 14.* **Visualizations of the domains we evaluate on.** Each of the 5 domains from OGBench has 5 tasks, for a total of 25 tasks.

global state-action information. The resulting sequence is processed by $L$ Transformer blocks with a pre-layer normalization architecture. Each block consists of multi-head self-attention followed by a two-layer feedforward network with GELU activations, with residual connections and dropout applied after both the attention and feedforward sublayers. Following common practice, we fix the head dimension, choosing $d_h = 32$ such that the number of heads $h$ is determined by $h = d_{\text{model}}/32$. The entropy computation of TQL can be done either after averaging attention scores over heads, or before averaging over heads and then averaging the entropies. In the experiments in this paper, the attention is averaged over heads in the layer before computing the entropy. After a final layer normalization, the representation corresponding to the [value] token is used as a global summary of the input. Finally, we predict an ensemble of $K$ Q-value estimates using $K$ lightweight MLP heads applied to the [value] embedding, producing $Q(s, a) \in \mathbb{R}^K$.

For the scaling experiments, we set $L = 2$ for the number of Transformer layers and vary the network size by setting the hidden dimension to $\{128, 256, 512, 1024\}$. Unless otherwise specified, we report the final performance of the largest model with hidden dimension 1024.

### C.3. Optimization Hyperparameters

In this section, we provide additional optimization and training details for TQL. All hyperparameters used in our experiments are summarized in Table 2. Following Park et al. (2025b), we use environment-specific BC coefficients $\alpha$ to account for differences in dataset quality and task difficulty. For the learning rate schedules, we adopt cosine decay for both the actor and critic. The actor learning rate is fixed to 5e-4, while the critic learning rate is set to 1e-4, which we found to be the best overall across dimensions, except for the largest model setting which we tune separately.

*Table 2.* Default training hyperparameters for TQL.

| Hyperparameter | Value |
| --- | --- |
| Optimizer | AdamW (weight decay 0.01) |
| Training steps | 1 M |
| Batch size | 256 |
| Discount factor ($\gamma$) | 0.99 |
| Target smoothing coefficient ($\tau$) | 0.005 |
| Actor learning rate | 5e-4 with cosine schedule |
| Critic learning rate | 1e-4 with cosine schedule |
| BC coefficient ($\alpha$) | Environment-specific, see Table 3 |
| Target attention entropy ($\bar{H}$) | Environment-specific, see Table 3 |
| Actor flow steps | 10 |
| Number of action samples | 16 |

**Attention Entropy Target.** The attention entropy target $\bar{H}$ is an important hyperparameter in TQL. The scale of attention entropy is primarily determined by the input dimensionality of the task, specifically the number of state dimensions $n_s$ and action dimensions $n_a$. In our experiments, we observe that the optimal target entropy depends on the environment being evaluated and is largely independent of the model size, but the entropy target should be tuned for each problem and setting for best performance.

To initialize $\bar{H}$ for each environment, we first compute a coarse upper bound on the attention entropy based on the input dimensionality. Concretely, the maximum entropy is

$$H_{\max} = \ln\left(1 + n_{\mathrm{s}} + n_{\mathrm{a}}\right), \tag{9}$$

and set the initial target entropy to $0.8 \times H_{\max}$. Starting from this value, we perform a local hyperparameter search within a $\pm 0.5$ range.

We further find that assigning smaller target entropy to the output layers leads to more stable training and encourages more deterministic predictions. We set a fixed $-0.5$ lower target entropy for the output layer in our experiments. All entropy tuning experiments are conducted using the second-largest model configuration, and the selected target entropy values are then fixed and reused for all model sizes within the same environment. The final target attention entropy values used for each environment are reported in Table 3.

### C.4. Baseline Implementation Details

We summarize the implementation details of all baseline methods used in our experiments. For BC, IQL, ReBRAC, FBRAC and IFQL we directly report the results from prior work (Park et al., 2025b; Dong et al., 2025d), which are averaged over 8 seeds. We report results of floq from their provided Weights and Biases logs, which are over 3 seeds. Domain-specific hyperparameters for all methods are summarized in Table 3. We report results for the ∼26M parameters network size setting for all transformer-based methods to ensure a fair comparison. Below, we provide additional implementation details for the remaining baselines.

**FQL (Park et al., 2025b).** For FQL, we report the results from the original paper for the main comparisons. For the scaling experiments, we train the critic with a fixed depth of four layers while varying the hidden dimension over {320, 512, 1536, 2944} to study the effect of model capacity. All hyperparameter settings are kept the same as in the official implementation.

**floq (Agrawalla et al., 2025).** For floq, we report performance at 1M training steps from the results provided in their released Weights and Biases logs, to ensure consistency in training steps with other baselines and our method. In the scaling experiments, we train the floq critic with four layers and vary the hidden dimension over {320, 512, 1536, 2944}. All hyperparameter settings are kept the same as in the official implementation.

**Q-Transformer (Q-T) (Chebotar et al., 2023).** For Q-Transformer, we use a causal Transformer decoder architecture as described in (Brohan et al., 2023). We implement QK Norm (Dehghani et al., 2023) to reduce attention collapse, and we use learned positional embeddings as the sequence length is fixed. With the exception of the causal mask, QK-Norm, and the input and output projections, this closely matches our baseline architecture, allowing for a direct comparison. We sweep over action bins, learning rate, AdamW weight decay, and the conservatism weight, finding an optimal $N = 64$, $lr = 2e - 5$, $\lambda = 0.02$, and $\alpha = 0.5$ (as defined in Eq. 2 of Chebotar et al. (2023)). We find that such comparatively low learning rates and high decay are necessary to achieve reasonable performance and otherwise observe significant instability. Moreover, without QK Norm we observe significant entropy collapse.

**Perceiver Actor-Critic (PAC) (Springenberg et al., 2024).** For PAC, we follow the critic architecture specified in the original paper, using 32 latent tokens and discretizing the value space into bins. A dense encoder is employed for both state and action representations. For value discretization, we set the lower bound to the minimum possible return and the upper bound to 0 for each task. The number of value bins is chosen such that each bin has width 1, corresponding to the minimum reward difference across all evaluated tasks. Although PAC is originally designed to jointly learn discrete action prediction, we find that this component leads to unstable training and worse performance in our evaluation setting. To ensure a fair comparison, we disable discrete action prediction and use the same policy extraction mechanism (Park et al., 2025b) as TQL to focus on value learning. For the scaling experiments, we train PAC models with two hidden layers and vary the hidden dimension over {80, 160, 320, 640}.

*Table 3.* **Domain specific hyperparameters for TQL and baselines. For attention entropy target** $\bar{H}$, the first dimension corresponds to Transformer layers. Within each tuple, the first value specifies the target entropy for the *value token*, while the second value applies to all other tokens.

| | IQL | ReBRAC | | FBRAC | IFQL | FQL | floq | PAC | TQL | |
|---|---|---|---|---|---|---|---|---|---|---|
| Domain or task | $\alpha$ | $\alpha_1$ | $\alpha_2$ | $\alpha$ | $N$ | $\alpha$ | $\alpha$ | $\alpha$ | $\alpha$ | $\bar{H}$ |
| cube-double-play | 0.3 | 0.1 | 0 | 100 | 32 | 300 | 300 | 300 | 300 | $((3.0, 3.0), (2.5, 2.5))$ |
| cube-triple-play | 10 | 0.03 | 0 | 100 | 32 | 300 | 300 | 300 | 300 | $((3.5, 3.5), (3.0, 3.0))$ |
| puzzle-3x3-play | 10 | 0.3 | 0.001 | 100 | 32 | 1000 | 1000 | 1000 | 1000 | $((3.5, 3.5), (3.0, 3.0))$ |
| puzzle-4x4-play | 3 | 0.3 | 0.01 | 300 | 32 | 1000 | 1000 | 1000 | 1000 | $((3.5, 3.5), (3.0, 3.0))$ |
| scene-play | 10 | 0.1 | 0.001 | 100 | 32 | 300 | 300 | 300 | 300 | $((3.5, 3.5), (3.0, 3.0))$ |

