# OpenReview forum: "TQL: Scaling Q-Functions with Transformers by Preventing Attention Collapse"
_ICML.cc/2026/Conference — ICML 2026 regular_

### Official Review · Reviewer_BJUc · 2026-03-12

**Soundness:** 3
**Presentation:** 4
**Significance:** 3
**Originality:** 3
**Overall Recommendation:** 4
**Confidence:** 4

**Summary:**

This paper focuses on scaling networks in Offline Deep Reinforcement Learning. Specifically, it concentrates on why naively scaling up transformer-based value networks hurts the performance. The authors argue that the attention collapse is the main reason for this. To tackle this issue, they propose to add an attention entropy loss to the objective function of value network optimization. The empirical experiments on the difficult OGBench demonstrate the effectiveness of the proposed Transformer Q-Learning (TQL) over other baselines floq and FQL.

**Compliance With Llm Reviewing Policy:**

Affirmed.

**Key Questions For Authors:**

1. Is it possible to expand the TQL to an online RL setting?
2. Is there any potential negative impact of an additional entropy loss?
3. Could the authors explain why entropy goes down, then goes up in Figure 3?

**Limitations:**

yes

**Strengths And Weaknesses:**

Strengths:
1. Scaling Deep RL is a significant and trending topic in the community.
2. The authors empirically diagnose the reason for the failure of scaling value functions and show that the attention collapse happens when they increase the critic model size.
3. The paper is well organized and presented.
4. The proposed solution is very simple, which makes it quite applicable to many methods.
5. The experimental results on OGBench are solid and persuasive.

Weaknesses
1. Since the authors focus on the niche of offline transformer-based Q-functions, the impact of this paper might be limited.
2. The discussion of the potential negative impact of the added entropy loss, i.e., limitation on the capacity of the network, is not conducted.

---

> ### Author Rebuttal · Authors · 2026-03-31
>
> Dear Reviewer BJUc
>
>
> Thank you for your thoughtful feedback! We provide clarifications to your questions below and will incorporate your feedback into the paper to enhance clarity.
>
>
> > Is it possible to extend to an online setting
>
> We evaluate TQL in an offline-to-online setting across cube-double, cube-triple, scene, and puzzle-4x4. TQL consistently outperforms both the vanilla transformer critic and strong baselines FQL and floq, achieving 3.8% higher average return. TQL scales significantly better than the transformer baseline without TQL, demonstrating its effectiveness beyond the purely offline regime. We will include additional online RL experiments in the camera-ready version.
>
>
> | Tasks | floq | FQL | TQL 0.4M | TQL 26M |
> |---|---|---|---|---|
> | cube-double | **39±27 → 99±2** | 40±11 → 92±3 | 75±27 → 73±31 | 84±8 → 96±5 |
> | puzzle-4x4 | 16±6 → 67±47 | 8±3 → 38±52 | 11±6 → 28±40 | **20±3 → 69±43** |
> | cube-triple | 10±4 → 82±24 | 4±1 → 83±12 | 23±10 → 51±23 | **29±16 → 96±5** |
> | scene | **90±11 → 100±0** | **82±11 → 100±1** | 39±13 → 81±28 | **81±14 → 100±0** |
> | **Average** | 38.8±12 → 87±18.3 | 33.5±6.5 → 78.3±17 | 37±14 → 58.3±30.5 | **53.5±10.3 → 90.3±13.3** |
>
>
>
>
> > Why does entropy go down then up in Figure 3
>
> In Figure 3, the entropy trajectory reflects two distinct phases of learning. In the early phase, attention heads specialize to focus on the most task-relevant input positions, which naturally reduces entropy. As training progresses, the model converges to a stable equilibrium where entropy settles at a moderate level, neither too high nor too low. In larger models, however, this self-correction fails and entropy collapses entirely and does not recover, which corresponds to the significantly degraded performance we observe at scale. This distinction motivates TQL's intervention to prevent collapse before it occurs.
>
>
> > Potential negative impact of an additional entropy loss
>
> The entropy loss is designed to guide attention toward a target range that prevents collapse while still permitting informative, specialized attention patterns. It does not, however, enforce uniformly high entropy and different heads can naturally develop different entropy levels, preserving the model's capacity for attention specialization. Empirically, the entropy loss improves performance across the board, including relative to non-transformer baselines that use no entropy regularization.
>
> The introduction of an entropy loss does introduce an additional hyperparameter. However, in our new sensitivity analysis below we find that TQL is not sensitive to the additional hyperparameter.
>
> **Experiments with different entropy targets on the cube double and puzzle 3×3 (0.4M model)**
> | Attention Entropy Target |  ((2.5, 2.5), (2.5, 2.5)) | ((3.0, 3.0), (2.5, 2.5)) | ((3.0, 3.0), (3.0, 3.0)) | ((3.5, 3.5), (3.0, 3.0)) |
> |---|---|---|---|---|
> | cube-double-task1 | 68±31 | 62±33 | 89±11 | 87±8 |
> | cube-double-task2 | 51±38 | 72±27 | 79±27 | 82±11 |
> | cube-double-task3 | 45±28 | 36±18 | 27±32 | 38±24 |
> | puzzle-3x3-task-1 | 86±12 | 84±9 | 85±8 | 76±9 |
> | puzzle-3x3-task-2 | 21±7 | 20±8 | 17±10 | 18±6 |
> | puzzle-3x3-task 3 | 20±5 | 12±4 | 10±3 | 13±7 |
> | **Average** | **48.5 ± 20.2** | **47.7 ± 16.5** | **51.2 ± 15.2** | **52.3 ± 10.8** |

---

> > ### Author Rebuttal · Reviewer_BJUc · 2026-04-02
> >
> > Thank you for the rebuttal.
> >
> > I appreciate the authors’ efforts. These additional experiments help extend the application of the proposed TQL and demonstrate the robustness of the hyperparameter choice in the entropy loss.
> >
> > I will maintain my positive rating of 4.

---

### Official Review · Reviewer_otB7 · 2026-03-13

**Soundness:** 4
**Presentation:** 3
**Significance:** 3
**Originality:** 3
**Overall Recommendation:** 5
**Confidence:** 4

**Summary:**

This paper investigates why scaling Transformer-based value functions in RL leads to performance degradation. Through empirical analysis, the authors identify attention collapse as the root cause: as model capacity grows, attention entropy decreases and weights concentrate on a few tokens, causing non-smooth Q-value landscapes and poor value estimates. To address this, the authors propose Transformer Q-Learning (TQL), which introduces per-layer learnable temperature parameters to guide attention entropy toward a target value during training, with a separate temperature for the [VALUE] token and learnable modality embeddings to distinguish state and action inputs. Evaluated on 25 tasks from OGBench, TQL achieves 43% improvement when scaling from 0.4M to 26M parameters while all baselines degrade, and achieves the best average success rate in fixed-scale comparisons against prior offline RL methods.

**Compliance With Llm Reviewing Policy:**

Affirmed.

**Final Justification:**

Most of my concerns have been addressed, I think this paper should be accepted

**Key Questions For Authors:**

see weakness

**Limitations:**

Yes

**Strengths And Weaknesses:**

## Strengths

**1.** The failure mode of Transformer value functions has been observed before but never systematically diagnosed. The authors support their attention collapse hypothesis through three complementary forms of evidence: attention entropy curves, Q-value landscape visualizations, and attention maps, consistently across five environments. This diagnosis has independent value beyond TQL itself and will remain useful for future work.

**2.** TQL adds only an entropy loss and a small set of temperature parameters with no architectural changes, making it directly applicable to any existing Transformer backbone including pretrained models. Compared to alternatives like network resets or mixture-of-experts, the integration cost is minimal. Notably, the key hyperparameter $\bar{H}$ takes nearly identical values across 4 out of 5 environments, confirming the method does not require careful per-environment tuning.

---

## Weaknesses

**1.** All experiments are in offline RL on a single benchmark. The paper claims TQL is a general framework, but this is untested in online RL where data distribution shifts continuously and plasticity issues are more prominent. Without at least one online RL experiment, the scope of the contribution is narrower than claimed.

**2.** The paper never examines the relationship between attention collapse and plasticity loss, which is a well-known failure mode in scaling RL networks. No plasticity metrics such as dead neurons or feature rank are measured. If attention collapse is a downstream symptom of plasticity loss rather than an independent cause, the mechanistic explanation offered by this paper would need to be revised.

---

> ### Author Rebuttal · Authors · 2026-03-31
>
> Dear Reviewer otB7
>
>
> Thank you for your insightful feedback and appreciation of our analysis and method! We provide clarifications to your questions below and will incorporate your feedback into the paper to enhance clarity.
>
>
>
> > Online RL
>
> We evaluate TQL in an offline-to-online setting across cube-double, cube-triple, scene, and puzzle-4x4. TQL consistently outperforms both the vanilla transformer critic and strong baselines FQL and floq, achieving 3.8% higher average return. TQL scales significantly better than the transformer baseline without TQL, demonstrating its effectiveness beyond the purely offline regime. We will include additional online RL experiments in the camera-ready version.
>
>
> | Tasks | floq | FQL | TQL 0.4M | TQL 26M |
> |---|---|---|---|---|
> | cube-double | **39±27 → 99±2** | 40±11 → 92±3 | 75±27 → 73±31 | 84±8 → 96±5 |
> | puzzle-4x4 | 16±6 → 67±47 | 8±3 → 38±52 | 11±6 → 28±40 | **20±3 → 69±43** |
> | cube-triple | 10±4 → 82±24 | 4±1 → 83±12 | 23±10 → 51±23 | **29±16 → 96±5** |
> | scene | **90±11 → 100±0** | **82±11 → 100±1** | 39±13 → 81±28 | **81±14 → 100±0** |
> | **Average** | 38.8±12 → 87±18.3 | 33.5±6.5 → 78.3±17 | 37±14 → 58.3±30.5 | **53.5±10.3 → 90.3±13.3** |
>
>
>
>
> > Using a plasticity loss
>
> Thank you for raising this important consideration. To isolate whether plasticity loss alone could address the observed degradation, we apply two popular plasticity losses, l2 init [1] and random resets [2], to the baseline transformer value function. All these variants perform significantly worse than TQL, indicating that plasticity loss does not resolve the underlying issue and is not the root cause of performance degradation. Furthermore, when inspecting the attention patterns of models trained with these plasticity loss corrections, we observe that attention entropy collapse persists, with attention weights remaining sharply concentrated on a small subset of tokens. This demonstrates that attention entropy collapse is a distinct failure mode that cannot be resolved by plasticity-aware training alone, and further supports the conclusion that attention entropy collapse is a distinct and primary failure mode that TQL specifically addresses.
>
> [1] Kumar et al., “Maintaining plasticity in continual learning via regenerative regularization,” arXiv preprint arXiv:2308.11958, 2023
> [2] Nikishin et al., “The Primacy Bias in Deep Reinforcement Learning,” arXiv:2205.07802, 2022.
>
>
> **Comparison with plasticity loss methods**
> | **Tasks** | Transformer w/o TQL & plasticity loss | Transformer w/ l2 init [1] | Transformer w/ random resets [2] | TQL |
> |---|---|---|---|---|
> | cube-double-task1 | 66±20 | 58±27 | 45±9 | **87±13** |
> | cube-double-task2 | 68±25 | 36±5 | **77±8** | 64±20 |
> | cube-double-task3 | 1±2 | 0±1 | 2±3 | **71±15** |
> | **Average** | 45 ± 15.7 | 31.3 ± 11 | 41.3 ± 6.7 | **74 ± 16** |

---

> > ### Author Rebuttal · Reviewer_otB7 · 2026-04-04
> >
> > Most of my concerns have been addressed, so I keep my high score

---

### Official Review · Reviewer_7iM4 · 2026-03-15

**Soundness:** 2
**Presentation:** 3
**Significance:** 3
**Originality:** 3
**Overall Recommendation:** 3
**Confidence:** 3

**Summary:**

The paper evaluates Transformer-based value functions on 25 tasks from OGBench and scales
them up to 26M parameters, improving success rates over prior offline RL methods. The authors
observe that low attention entropy correlates with poor performance and rugged value landscapes.
They introduce attention entropy regularization, which stabilizes attention entropy, smooths value
landscapes, and improves success rates when scaling value functions.

**Compliance With Llm Reviewing Policy:**

Affirmed.

**Final Justification:**

We thank the authors for the detailed responses.

While most of our questions were clarified, which leads us to raise the presentation score to 3, the concern regarding environment choices remains unresolved.

Although we agree with the authors on the number of experiments and cube-triple, the exclusion of five environments from the benchmark used in prior work (e.g., FQL and floq) is still unexplained.

Not addressing this exclusion can significantly affect the soundness and significance of the work. Consequently, our overall score remains unchanged.

**Key Questions For Authors:**

1. Why was the proposed entropy regularization formulation chosen instead of a simpler squared-
error penalty between actual and target entropy? Was this design choice empirically validated
over others? The dual-variable terms scheme appears to converge reliably in the reported
tasks. Can the authors provide intuition or conditions under which this convergence occurs?
A satisfactory response could improve my assessment of soundness and presentation.
2. Why were the experiments conducted only on a subset of the benchmark used in prior methods
(e.g. floq and Flow Q-Learning)? Do the authors expect the observed effects to generalize to
the remaining environments? Any qualitative insights or preliminary results would be helpful.
A clear response could improve my assessment of soundness, presentation and significance.
3. Can the authors include uncertainty measures (e.g., standard deviations) for the relative
improvement figures (Figures 1 and 4)?
A satisfactory response, either with figures or tables, would moderately improve my assess-
ment of soundness.
4. Are the architectures and parameter counts of the Flow Q-learning policy extraction models
fixed across scaling experiments?
A satisfactory response may improve my assessment of presentation.
5. The paper focuses on Q-learning. Can the authors comment on whether attention entropy
regularization could similarly stabilize learning in other RL paradigms, and what challenges
might arise?
A satisfactory response may improve my assessment of significance.

**Limitations:**

yes

**Strengths And Weaknesses:**

Soundness

Strengths

• The paper presents empirical evidence, over 25 tasks from OGBench, supporting the claim
that attention entropy regularization improves success rates when scaling Transformer-based
value functions.
• The paper identifies attention entropy collapse as a main bottleneck.
• The method does not require structural changes to the Transformer.
• Fair comparisons with prior offline RL methods using the same policy extraction scheme.
• Comparisons with other transformer stabilization methods are provided (Appendix).
• Ablations isolate components of the loss.

Weaknesses

1. The authors chose a particular subset of the OGBench single-task variants (manipulation
env.) as their evaluation suite, without explaining why, while prior methods such as floq and
Flow Q-Learning were both evaluated on the same 50 (single-task variants for locomotion
and manipulation env.) tasks from OGBench.
2. Table 1 reports TQL results over 3 seeds, whereas OGBench typically uses 4–8 seeds. Using
the standard benchmark protocol would strengthen the comparison.
3. Relative improvement figures (Figures 1 and 4) could benefit from uncertainty estimates since
the results from Figure 1 are directly reported in the abstract.
4. No ablation is provided comparing the proposed regularization to simpler formulations that
directly penalize deviation from a target entropy.

Furthermore, several secondary/less important claims are supported only by visual inspection
of a small number of plots without aggregated metrics or uncertainty estimates.

1. The paper claims that entropy collapse correlates with non-smooth value landscapes and poor
task performance (lines 366–367, 424–428 (2nd column), 670–671), but no correlation analysis
is provided.

2. The smoothness of the value landscapes is only assessed visually. No quantitative metric is
reported.

3. Attention entropy plots do not report variability (e.g., standard deviation).
Overall: Because of these weaknesses, the soundness is rated as fair. This could improve with
additional explanations, analyses, and metrics.

Presentation

Strengths
• Mostly understandable, but explanations are sometimes minimal.
• Figures are clear and readable.
• Appendix includes value function architecture and hyperparameters.

Weaknesses

Clarifications would improve the paper:

1. The dual-variable regularization terms scheme is not clearly explained or motivated compared
to other methods to guide entropy towards a target value.
2. The architecture and hyperparameters used for policy extraction (Flow Q-learning) are not
restated, which harms reproducibility.
3. The abstract, early sections, and captions (e.g. Figure 1) should explicitly state that the
evaluation suite is from a subset of OGBench and why they chose it over using the same
benchmark as in prior methods like floq and Flow Q-Learning.
4. The discussion of attention entropy control should reference prior work from the NLP lit-
erature more clearly in the main text (some are only mentioned in Appendix B.2). The
work should clearly clarify/distinguish what is new from what is not, as attention entropy
regularization in NLP is not novel.
5. Some notation around the Flow Q-learning equations (Eq. 6–8) requires reading the original
paper.
6. The description of Q-value landscape visualizations is insufficient. It is unclear how the
ensemble of Q-values is aggregated into a single value for visualization.

Two points are particularly important:

• Explanation of the regularization choice.
• Details of the policy extraction architecture and hyperparameters.

Significance

1. Attention entropy regularization improves performance and provides insights into instability
causes.
2. The approach could influence future work on scaling RL models and applying transformer
stabilization techniques from NLP.
3. However, only a subset of the benchmark tasks used in prior studies is evaluated, which may
weaken the demonstrated applicability and significance.
4. Furthermore, the paper only focuses on Q-learning which can decrease the scope of impact.

Originality

1. Regularizing attention entropy is not entirely new, but its application to Transformer-based
value functions in offline RL is novel.
2. More discussion of related NLP stabilization methods (Appendix B.2) in the main text would
strengthen context.

---

> ### Author Rebuttal · Authors · 2026-03-31
>
> Dear Reviewer 7iM4
>
> Thank you for your thoughtful feedback, we address the comments below and will incorporate your feedback into the final version of the paper.
>
>
> > Environment choices for experiments
>
> Unlike prior works that focus on performance given a fixed model size, our experiments focus on scaling with a range of model sizes, and whether TQL can enable value functions to scale with a transformer.
>
> To this end, we choose 5 domains from OGBench, each containing 5 tasks, for a total of 25 tasks to rigorously analyze the effects of scaling behavior over 4 model sizes, which involves 2x the number of experiments compared to baselines.
>
> We also highlight that our evaluation includes the cube-triple benchmark, which is an especially challenging task that FQL and floq omit, further demonstrating TQL's effectiveness in challenging settings.
>
>
> > Proposed entropy regularization formulation chosen
>
> The entropy regularization formulation is a central design choice of TQL. In Figure 5, we compare against alternative formulations, including max entropy regularization, which performs significantly worse.
>
> Per the suggestion of the reviewer, we evaluate a naive squared-error penalty between actual and target entropy, and find that it similarly underperforms in both performance and scaling. This is because entropy collapses during training and a max entropy objective can recover it toward a baseline level. However, naively applying max entropy is difficult to balance in practice.
>
> **Ablation study of L2 loss entropy tuning on the 26M transformer**
> | Task | l2 loss entropy tuning | TQL |
> |---|---|---|
> | cube-double-task1 | 67±12 | **95±2** |
> | cube-double-task2 | **88±5** | 84±8 |
> | cube-double-task3 | 15±18 | **60±27** |
> | cube-double-task4 | **14±9** | **14±6** |
> | cube-double-task5 | 24±15 | **84±9** |
> | **Average** | 41.6 ± 11.8 | **67.4 ± 10.4** |
>
> We therefore design the adaptive temperature to control entropy so that it does not fall below the target which provides a stable, directed correction rather than unconstrained maximization. We will more clearly motivate this design in the camera-ready version.
>
>
> > Can the authors include uncertainty measures
>
> Thank you for bringing this up. Uncertainty measures are present in the OGBench evaluation results in Table 1, from which the relative improvement plots (Figure 1) are directly derived. We will ensure that uncertainty measures are included directly in the relative improvement plots in the camera-ready version as well.
>
>
>
> > The architecture and hyperparameters used for policy extraction (Flow Q-learning) are not restated
>
> We follow the design choices from FQL for policy extraction. Specifically, we use two MLP-based flow matching networks, each with hidden dimensions (512, 512, 512, 512): one for the multi-step behavior cloning model and one for the one-step distilled policy, which is trained using a distillation objective from the multi-step model combined with a Q-maximization loss. The default number of function evaluation steps for the multi-step behavior cloning model is set to ten. We will restate these details explicitly in the main paper to improve reproducibility.
>
>
> > Architectures and parameter counts of the Flow Q-learning policy extraction models across scaling experiments
>
> We apologize for the confusion. The architecture and parameter counts for the policy extraction models (actors) are fixed across all scaling experiments.
>
> To clearly demonstrate the effect of our method, we only vary the critic network size by scaling the hidden dimension as described in Section 5.3 and further Appendix C.2. We will make this distinction explicit in the final version.
>
>
> > Other paradigms besides Q-Learning
> Our work focuses on Q-learning as a foundational paradigm in value-based deep RL, and our experiments are designed accordingly. That said, we expect TQL's benefits to generalize to other settings where a transformer is used as a value function.
>
> To demonstrate the impact of our method beyond offline RL, we also report results in an offline-to-online RL setting, where the model is trained offline for one million steps followed by one million steps of online training. TQL consistently outperforms baselines and scales effectively in this setting, suggesting TQL is also effective in online RL settings without change.
>
> | Tasks | floq | FQL | TQL 0.4M | TQL 26M |
> |---|---|---|---|---|
> | cube-double | **39±27 → 99±2** | 40±11 → 92±3 | 75±27 → 73±31 | 84±8 → 96±5 |
> | puzzle-4x4 | 16±6 → 67±47 | 8±3 → 38±52 | 11±6 → 28±40 | **20±3 → 69±43** |
> | cube-triple | 10±4 → 82±24 | 4±1 → 83±12 | 23±10 → 51±23 | **29±16 → 96±5** |
> | scene | **90±11 → 100±0** | **82±11 → 100±1** | 39±13 → 81±28 | **81±14 → 100±0** |
> | **Average** | 38.8±12 → 87±18.3 | 33.5±6.5 → 78.3±17 | 37±14 → 58.3±30.5 | **53.5±10.3 → 90.3±13.3** |

---

> > ### Author Rebuttal · Reviewer_7iM4 · 2026-04-07
> >
> > We thank the authors for the detailed responses.
> >
> > While most of our questions were clarified, which leads us to raise the presentation score to 3, the concern regarding environment choices remains unresolved.
> >
> > Although we agree with the authors on the number of experiments and cube-triple, the exclusion of five environments from the benchmark used in prior work (e.g., FQL and floq) is still unexplained.
> >
> > Not addressing this exclusion can significantly affect the soundness and significance of the work. Consequently, our overall score remains unchanged.

---

> > > ### Author Response · Authors · 2026-04-08
> > >
> > > Thank you for raising the presentation score and for the continued engagement. We are happy to hear that we were able to clarify your concerns besides environment choices.
> > >
> > > Regarding environment choices, we note that because our experiments focus on scaling, we end up running substantially more experiments than FQL and floq. Specifically, each algorithm including TQL and baselines require 25 tasks × 3 seeds = 75 runs.
> > >
> > > For the scaling experiment, we train 4 algorithms on these tasks, namely, TQL, FQL, floq, and PAC.
> > >
> > > For the largest model, each run takes about 36 hours on a single A5000 GPU to train for 1M steps. This results in a total of 75 × 4 × 36 hours = 450 days of compute on a single A5000 GPU.
> > >
> > > In comparison to floq, floq runs on 50 tasks × 3 seeds = 150 runs. Each run takes about 4 hours on a single A5000 GPU. The total compute required is 150 × 4 hours = 25 days on a single A5000 GPU.
> > >
> > > Overall, our experiments require 450 / 25 = 18× more compute time.
> > >
> > > Given these constraints in an academic setting, we selected 25 tasks from OGBench to focus the scaling experiments. However, as per the suggestion of the reviewer, we will include the additional environments in the camera ready version of the paper. We hope the above breakdown clarifies the number of environments, and we kindly invite the reviewer to reconsider their overall score in light of this.

---

### Official Review · Reviewer_mjYp · 2026-03-16

**Soundness:** 3
**Presentation:** 3
**Significance:** 3
**Originality:** 3
**Overall Recommendation:** 4
**Confidence:** 3

**Summary:**

This paper studies why transformer architectures fail to scale when used as value functions in reinforcement learning. While transformers have demonstrated strong scaling behavior in domains such as NLP and vision, the authors observe that increasing the capacity of transformer-based Q-functions leads to worse performance and training instability. Through empirical analysis, the paper identifies attention entropy collapse as the main failure mode: as model size increases, attention distributions become highly concentrated on a few tokens, preventing effective learning of value estimates. This collapse produces unstable and non-smooth Q-value landscapes and significantly degrades task performance. To address this issue, the authors propose Transformer Q-Learning (TQL), which introduces a mechanism to control the entropy of attention distributions during training. The method adds entropy regularization and a learnable temperature parameter to guide attention entropy toward a target value. This prevents attention collapse and stabilizes value learning as model size increases. The method is evaluated on the OGBench offline RL benchmark across 25 tasks and several model scales. Results show that TQL enables consistent scaling of transformer-based value functions and improves performance compared to several offline RL baselines, achieving the best average performance across tasks.

**Compliance With Llm Reviewing Policy:**

Affirmed.

**Key Questions For Authors:**

1. Why does attention entropy collapse occur specifically in transformer-based value functions? Can the authors provide a deeper explanation or theoretical insight into this phenomenon?
2. How sensitive is the method to the target attention entropy hyperparameter? Does performance vary significantly with different entropy targets?
3. Would TQL also improve stability in online RL settings? For example, when training critics in actor–critic methods.
4. Could enforcing higher attention entropy limit the model’s ability to focus on important tokens?
5. How does the method balance preventing collapse with allowing meaningful attention specialization?
6. How does TQL perform with significantly larger models (e.g., >100M parameters)?
7. Can the entropy control mechanism be applied to other transformer-based RL models, such as decision transformers or world models?
8. How does TQL compare computationally with baseline transformer critics in terms of training overhead?
9. Have the authors considered evaluating TQL in pixel-based environments or with model-free algorithms such as Rainbow? It would be interesting to understand whether the attention entropy collapse phenomenon also appears in these settings and whether TQL provides similar benefits.

**Limitations:**

Yes!

**Strengths And Weaknesses:**

**Strengths**
1. The paper addresses the challenge of scaling value functions in reinforcement learning, an area where larger models often lead to instability rather than improved performance.
2. The authors identify attention entropy collapse as a key failure mode when scaling transformer-based Q-functions, supported by attention entropy measurements and Q-value landscape visualizations.
3. The proposed Transformer Q-Learning (TQL) introduces a lightweight mechanism to guide attention entropy toward a target value, requiring minimal architectural changes.
4. The method is evaluated on 25 tasks across 5 domains in OGBench, showing consistent improvements over multiple offline RL baselines.
5. Unlike prior methods whose performance degrades with increased model size, TQL improves performance as model capacity increases, demonstrating meaningful scaling.
6. The paper includes ablations showing the impact of removing entropy guidance, using fixed entropy penalties and using a single temperature parameter.
7. Figures showing attention entropy trends, Q-value landscapes, and scaling curves help illustrate the proposed failure mode and the benefits of the method.

**Weaknesses**
1. The paper provides empirical evidence for attention collapse but lacks theoretical analysis explaining why attention entropy collapses in value learning settings.
2.  All experiments are conducted in offline RL environments. It is unclear whether the method would also be effective in online RL settings.
Moderate novelty of the method: The core idea—entropy regularization with temperature tuning for attention—is conceptually related to existing entropy regularization techniques.
3. The method introduces a target entropy hyperparameter, and the sensitivity of performance to this parameter is not thoroughly analyzed.
4. The evaluation focuses on OGBench, and results on other commonly used benchmarks (e.g., D4RL or Atari) would strengthen the empirical claims.
5. The largest model evaluated contains 26M parameters, which is still relatively small compared to modern transformer scaling regimes.
6. The paper does not explore whether the proposed entropy control mechanism would benefit transformer-based policies or other RL architectures.

---

> ### Author Rebuttal · Authors · 2026-03-31
>
> Dear Reviewer mjYp,
>
> Thank you for your thorough and constructive review! We provide clarifications to your questions below, which will be incorporated into the paper to enhance clarity.
>
> >  Understanding attention entropy collapse
>
> We provide an intuitive understanding of why attention collapses in value learning settings. In supervised learning, dense and stationary gradient signal naturally prevents attention from degenerating. In Q-function learning, targets are bootstrapped and non-stationary — particularly early in training, which includes more bias and high-variance gradients. Under these conditions, the attention mechanism reduces variance by collapsing toward degenerate distributions, which then destabilizes Q-value estimates. This is consistent with our visualizations: entropy collapse consistently precedes Q-value deterioration, suggesting it is a cause rather than a symptom. Formalizing this intuition via gradient dynamics analysis under bootstrapped targets is a compelling direction we will highlight as future work in the revision.
>
> Regarding Q4 and Q5 — The adaptive temperature mechanism guides entropy toward a target that still allows for informative attention while preventing collapse. As we discuss in Section A, we empirically find that the addition of the target entropy does not produce flat Q-value landscapes and allows the attention to specialize. This is confirmed by our Q-value landscape visualizations with TQL having smoother, more structured landscapes than baselines instead of flatter ones. We also note that the target entropy term only limits the attention entropy at one layer and there are multiple layers, each with multiple heads, so the network is still able to concentrate on specific tokens, even though any given head is disincentivized from entirely attending to a specific token. Prior such as [1] find similar results to ours in the context of language modeling, finding that collapsed entropy regimes result in highly degraded performance, with the best results obtained within entropy regimes that resemble our tuning heuristic.
>
> [1] J. Hong and S. Lee. Variance Sensitivity Induces Attention Entropy Collapse and Instability in Transformers. EMNLP, 2025.
>
>
>
>
>
>
>
>
>
> > Generalization beyond offline RL
>
> We evaluate TQL in an offline-to-online setting on cube-double, cube-triple, scene, and puzzle-4x4. TQL consistently outperforms the vanilla transformer critic and strong baselines FQL and floq, achieving 3.8% higher average return. Notably, TQL also scales significantly better than the transformer baseline without TQL, demonstrating its effectiveness in online settings as well. We will include additional online RL experiments in the camera-ready version.
>
> | Tasks | floq | FQL | TQL 0.4M | TQL 26M |
> |---|---|---|---|---|
> | cube-double | **39±27 → 99±2** | 40±11 → 92±3 | 75±27 → 73±31 | 84±8 → 96±5 |
> | puzzle-4x4 | 16±6 → 67±47 | 8±3 → 38±52 | 11±6 → 28±40 | **20±3 → 69±43** |
> | cube-triple | 10±4 → 82±24 | 4±1 → 83±12 | 23±10 → 51±23 | **29±16 → 96±5** |
> | scene | **90±11 → 100±0** | **82±11 → 100±1** | 39±13 → 81±28 | **81±14 → 100±0** |
> | **Average** | 38.8±12 → 87±18.3 | 33.5±6.5 → 78.3±17 | 37±14 → 58.3±30.5 | **53.5±10.3 → 90.3±13.3** |
>
>
>
>
>
> > Robustness and sensitivity
>
> We report results for analyzing the impact of target entropy on performance. We see that TQL is generally robust to the hyperparameter of target entropy. This demonstrates that TQL does not require careful per-task tuning and that the target entropy can be set once and transferred across tasks. We will include this analysis in the camera-ready version.
>
> **Experiments with different entropy targets on the cube double and puzzle 3×3 (0.4M model)**
> | Attention Entropy Target |  ((2.5, 2.5), (2.5, 2.5)) | ((3.0, 3.0), (2.5, 2.5)) | ((3.0, 3.0), (3.0, 3.0)) | ((3.5, 3.5), (3.0, 3.0)) |
> |---|---|---|---|---|
> | cube-double-task1 | 68±31 | 62±33 | 89±11 | 87±8 |
> | cube-double-task2 | 51±38 | 72±27 | 79±27 | 82±11 |
> | cube-double-task3 | 45±28 | 36±18 | 27±32 | 38±24 |
> | puzzle-3x3-task-1 | 86±12 | 84±9 | 85±8 | 76±9 |
> | puzzle-3x3-task-2 | 21±7 | 20±8 | 17±10 | 18±6 |
> | puzzle-3x3-task 3 | 20±5 | 12±4 | 10±3 | 13±7 |
> | **Average** | **48.5 ± 20.2** | **47.7 ± 16.5** | **51.2 ± 15.2** | **52.3 ± 10.8** |
>
>
>
> > Computational Cost and Practicality
>
> TQL's entropy guidance adds minimal overhead as it requires computing mean attention entropy per layer which is dominated by the existing O(n²) attention computation and updating a scalar temperature via a lightweight update. We report wall clock time of TQL. We see that TQL incurred only a 1.4% increase in wall-clock training time relative to the vanilla transformer baseline with 26M parameters. This confirms that TQL's benefits come essentially at no additional computational cost.
>
> | | One update (on L40 GPU) |
> |---|---|
> | **Transformer w/o TQL** | 0.07 sec |
> | **TQL** | 0.071 sec |

---

> > ### Author Rebuttal · Reviewer_mjYp · 2026-04-04
> >
> > Thank you for addressing my concerns. I appreciate the authors’ efforts. I will maintain my positive rating of 4. I suggest sharing the scripts to ensure reproducibility of the main experiments in the paper and the offline-to-online setting. Could you please provide them via an anonymous repository?
> >
> > >“We will include additional online RL experiments in the camera-ready version.”
> >
> > Q1. Which setup are you considering for this setting? Please explain why this setup is appropriate.
> >
> > >“TQL consistently outperforms the vanilla transformer critic and strong baselines FQL and floq.”
> >
> > I consider this statement somewhat strong. Comparing scaled TQL with FQL/floq may not be appropriate, as FQL/floq are not scaled (i.e., do not have the same number of parameters), unless this is the case?

---

> > > ### Author Response · Authors · 2026-04-08
> > >
> > > Thank you for your reply!
> > >
> > > > Which setup are you considering for this setting?
> > >
> > > We will include additional online RL experiments on the OGBench environments in the camera-ready version, similar to the experimental protocols in the offline setting. We believe this is a natural and appropriate choice, as it allows for a direct, controlled comparison to other offline RL methods that show offline-to-online results on OGBench such as FQL and floq.
> > >
> > >
> > > > Comparing scaled TQL with FQL/floq may not be appropriate, as FQL/floq are not scaled
> > >
> > > An important aspect of TQL is its ability to scale effectively with the number of parameters, which is a property that, as demonstrated in the scaling experiments and Figure 4, is absent in FQL/floq.  We report both the small and large scales for offline to online experiments to show that (1) TQL's performance improves with scale, and (2) this advantage is maintained in the offline-to-online setting. We will clarify this framing when putting the offline-to-online experiments in the revised manuscript.
> > >
> > >
> > > > I suggest sharing the scripts to ensure reproducibility of the main experiments
> > >
> > > We have uploaded an anonymous repository containing scripts to reproduce the main experiments, including the offline-to-online setting: https://anonymous.4open.science/r/tql-rebuttal-24CC/README.md.

---

### Decision · Program_Chairs · 2026-04-30

**Decision:**

Accept (regular)

**Comment:**

This submission was received positively overall. Reviewers found the paper to address an important and timely problem in reinforcement learning, namely the difficulty of scaling transformer-based value functions. The paper was praised for identifying a compelling empirical failure mode—attention entropy collapse—and for proposing a simple and effective intervention, Transformer Q-Learning (TQL), that substantially improves scaling behavior without requiring architectural changes. Reviewers also highlighted the strength of the experimental study, including the breadth of the evaluation across OGBench tasks, the clarity of the empirical diagnosis, and the practical appeal of the method.

Several limitations were noted during review, including the lack of theoretical analysis of the collapse phenomenon, the focus on offline RL, and the absence of broader validation on additional benchmarks or larger-scale regimes. However, these concerns were either substantially mitigated or judged acceptable in light of the paper’s scope and contributions, and the rebuttal was generally viewed as strong and responsive.

The main remaining point of concern relates to the choice of evaluation environments. One reviewer maintained reservations about the exclusion of some environments used in prior OGBench-based work. I appreciate this concern, but I also find the authors’ response compelling. In particular, the paper’s goal is not merely to report performance at a fixed model size, but to study scaling behavior across multiple model sizes and methods, which substantially increases the experimental burden relative to prior work. In that context, the choice to evaluate on 25 tasks appears well justified, especially given the already considerable computational scope of the study and the breadth of the resulting analysis. I therefore do not view this issue as undermining the soundness or significance of the contribution.

On balance, I recommend acceptance. The paper provides a meaningful empirical contribution, introduces a practical and lightweight method, and advances understanding of scaling behavior in transformer-based value learning. I encourage the authors to incorporate the reviewers’ suggestions in the final version, especially regarding the discussion of scope and limitations.